# Irp2 regulates insulin production through iron-mediated Cdkal1-catalyzed tRNA modification

Maria C. Ferreira dos Santos [1,2,18], Cole P. Anderson[2,3,12,18], Susanne Neschen[4,5,18], Kimberly B. Zumbrennen-Bullough[2,3], Steven J. Romney[1,2,13], Melanie Kahle-Stephan[4,14], Birgit Rathkolb [4,5,6], Valerie Gailus-Durner [4], Helmut Fuchs [4], Eckhard Wolf [6], Jan Rozman [4,5,15], Martin Hrabe de Angelis [4,5,7], Weiling Maggie Cai[8,9,16], Malini Rajan [1,2], Jennifer Hu[10,17], Peter C. Dedon [9,11] & Elizabeth A. Leibold [1,2,3]*

Regulation of cellular iron homeostasis is crucial as both iron excess and deficiency cause hematological and neurodegenerative diseases. Here we show that mice lacking iron-regulatory protein 2 (Irp2), a regulator of cellular iron homeostasis, develop diabetes. Irp2 post-transcriptionally regulates the iron-uptake protein transferrin receptor 1 (TfR1) and the iron-storage protein ferritin, and dysregulation of these proteins due to Irp2 loss causes functional iron deficiency in β cells. This impairs Fe–S cluster biosynthesis, reducing the function of Cdkal1, an Fe–S cluster enzyme that catalyzes methylthiolation of $t^6A37$ in $tRNA^{Lys}_{UUU}$ to $ms^2t^6A37$. As a consequence, lysine codons in proinsulin are misread and proinsulin processing is impaired, reducing insulin content and secretion. Iron normalizes $ms^2t^6A37$ and proinsulin lysine incorporation, restoring insulin content and secretion in $Irp2^{-/-}$ β cells. These studies reveal a previously unidentified link between insulin processing and cellular iron deficiency that may have relevance to type 2 diabetes in humans.

[1] Department of Medicine, Division of Hematology, University of Utah, Salt Lake City, UT 84112, USA. [2] Molecular Medicine Program, University of Utah, Salt Lake City, UT 84112, USA. [3] Department of Oncological Sciences, University of Utah, Salt Lake City, UT 84112, USA. [4] German Mouse Clinic, Institute of Experimental Genetics, Helmholtz Zentrum München, Ingolstädter Landstraße 1, 85764 Neuherberg, Germany. [5] German Center for Diabetes Research (DZD), Ingolstädter Landstraße 1, 85764 Neuherberg, Germany. [6] Institute of Molecular Animal Breeding and Biotechnology, Gene Center, Ludwig-Maximilians-Universität München, Feodor-Lynen Strasse 25, 81377 Munich, Germany. [7] Chair of Experimental Genetics, School of Life Science Weihenstephan, Technische Universität München, Alte Akademie 8, 85354 Freising, Germany. [8] Department of Microbiology, National University of Singapore, Singapore, Singapore 119077. [9] Antimicrobial Resistance Interdisciplinary Research Group (IRG), Singapore-MIT Alliance for Research and Technology, 1 CREATE Way, Singapore, Singapore 138602. [10] Department of Chemistry, Massachusetts Institute of Technology, Cambridge, MA 02139, USA. [11] Department of Biological Engineering, Massachusetts Institute of Technology, Cambridge, MA 02139, USA. [12] Present address: Landstuhl Regional Medical Center, 66849 Landstuhl, Germany. [13] Present address: Thermo Fisher Scientific, Waltham, MA 02451, USA. [14] Present address: Medizinische Hochschule Brandenburg Theodor Fontane Institut für Sozialmedizin und Epidemiologie, 14770 Brandenburg an der Havel, Germany. [15] Present address: Czech Centre for Phenogenomics, Institute of Molecular Genetics of the Czech Academy of Sciences, Prumyslova 595, 252 50 Vestec, Czech Republic. [16] Present address: Agilent Technologies, 1 Yishun Ave 7, Singapore, Singapore 768923. [17] Present address: Celgene Corporation, 1616 Eastlake Ave East, Seattle, WA 98102, USA. [18] These authors contributed equally: Maria C. Ferreira dos Santos, Cole P. Anderson, Susanne Neschen. *email: betty.leibold@genetics.utah.edu

Diabetes is characterized by high blood glucose levels due to the inability of pancreatic β cells to produce sufficient insulin to meet the needs of the body. Accumulating evidence reveals a role for iron in the pathogenesis of diabetes. Excess body iron stores are associated with an increased risk of type 2 diabetes (T2D) in patients with the genetic iron-overload disease hemochromatosis and in healthy individuals[1–3]. Phlebotomy improves insulin sensitivity, insulin secretion, and glucose regulation in these individuals showing a causal role of iron in T2D[1]. The mechanisms through which excess iron contributes to development of T2D are not yet fully understood but likely involves insulin resistance as well as impaired pancreatic β-cell function. Iron is important for normal glucose-stimulated insulin secretion[4]; however, excess iron causes oxidative stress and increases apoptosis in β cells[4–6]. While the consequences of excess iron in β-cell function and survival are established, the effects of iron deficiency on β-cell function and diabetes risk in humans are not yet fully understood.

Eukaryotic cells require iron for survival due to its presence in proteins involved in key cellular processes. Iron is used for the mitochondrial synthesis of Fe–S clusters and heme, which are cofactors for electron transport complexes and tricarboxylic acid (TCA) cycle enzymes and for the maturation of extra-mitochondrial Fe–S proteins involved in DNA metabolism, translation, and tRNA modification[7–9]. Precise regulation of cellular iron content is crucial as excess iron generates reactive oxygen species (ROS) that damage DNA and proteins, while cellular iron deficiency causes mitochondrial dysfunction and cell cycle arrest. All organisms have thus developed mechanisms to sense, acquire, and store iron.

In vertebrates, cellular iron homeostasis is controlled post-transcriptionally by iron-regulatory protein 1 (Irp1, Aco1) and iron-regulatory protein 2 (Irp2, Ireb2)[10–12]. Irps bind to RNA stem-loops known as iron-responsive elements (IREs) in the 5′- or 3′-untranslated regions (UTRs) of mRNAs involved in iron uptake (transferrin receptor 1; TfR1, Tfrc) and sequestration (ferritin-H and -L subunits; Fth1, Ftl1), and regulate the stability or translation of these mRNAs. When cells are iron deficient, Irps bind IREs with high affinity, inhibiting the translation of ferritin mRNA, while stabilizing TfR1 mRNA. When cells are iron-sufficient, Irps bind with low affinity to IREs, increasing ferritin synthesis, and promoting TfR1 mRNA degradation. Iron regulates Irp1 and Irp2 by different post-transcriptional mechanisms: Irp1 assembles a Fe–S cluster and is converted to cytosolic aconitase lacking RNA-binding activity, while Irp2 is targeted for proteasomal degradation by the E3 ubiquitin ligase Fbxl5[13,14]. By sensing cellular iron concentration, Irps regulate the amount of iron acquired by TfR1 and sequestered by ferritin, thus maintaining cellular iron within a narrow range to avoid adverse consequences of iron excess or depletion.

Our studies as well as others reported that $Irp2^{-/-}$ mice develop microcytic anemia[15–17] and adult-onset neurodegenerative disease[17,18] caused by dysregulation of iron homeostasis and functional iron deficiency in erythroid precursor cells and in neurons, respectively. Here, we show that $Irp2^{-/-}$ mice develop diabetes as a consequence of functional iron deficiency in pancreatic β cells. We found that cellular iron deficiency impairs the function of Cdk5-regulatory subunit-associated protein 1-like 1 (Cdkal1), a radical S-adenosylmethionine (SAM) enzyme that contains 4Fe–4S clusters required for its methylthiotransferase activity. Cdkal1 catalyzes the methylthiolation of $N^6$-threonylcarbamoyl adenosine 37 (t6A37) in cytosolic tRNA$^{Lys}_{UUU}$ to generate ms2t6A37, which is required for the accurate translation of lysine codons in proinsulin[19,20]. As a consequence of reduced Cdkal1 function in $Irp2^{-/-}$ β cells,

lysine codons in proinsulin are misread and proinsulin processing is impaired, reducing insulin content and secretion. The ms2t6A37 modification in tRNA$^{Lys}_{UUU}$ is also relevant to humans as genome-wide association studies showed CDKAL1 as a strong T2D susceptibility gene[21–24]. Our studies show a critical role for Irp2 in the regulation of β cell iron homeostasis and reveal a previously unrecognized role for iron in proinsulin processing and insulin secretion in these cells.

## Results

**$Irp2^{-/-}$ mice develop glucose intolerance.** While performing neurological analysis of aged $Irp2^{-/-}$ mice[17], we found that plasma glucose concentrations were elevated in both random-fed and fasted 12-month-old male $Irp2^{-/-}$ mice compared with WT (random-fed: WT, $150 \pm 13$ mg/dl versus $Irp2^{-/-}$, $211.2 \pm 13.6$ mg/dl glucose, $n = 9$–10, $p < 0.01$, and fasted: WT, $95.0 \pm 5.3$ mg/dl versus $Irp2^{-/-}$, $118 \pm 7.1$ mg/dl glucose, $n = 7$–10, $p < 0.05$). Glucose concentration also showed a tendency to be elevated in random-fed 3-month-old male $Irp2^{-/-}$ mice, and significantly elevated in female $Irp2^{-/-}$ mice (Supplementary Table 1). The observed changes in glucose homeostasis in 3-month-old mice were not attributable to alterations in lipid homeostasis as plasma total cholesterol, triglycerides, nonesterified fatty acids, and electrolytes were similar in WT and $Irp2^{-/-}$ mice (Supplementary Table 1). Similar to aged $Irp2^{-/-}$ mice[17], male and female 3-month-old mice showed mild microcytic anemia characterized by reduced hemoglobin and hematocrit and decreased erythrocyte indices (Supplementary Table 2).

To assess whole-body glucose metabolism, intraperitoneal glucose tolerance tests (ipGTTs) were performed on overnight fasted male WT and $Irp2^{-/-}$ mice. After glucose injection, $Irp2^{-/-}$ mice at 5-months of age exhibited an increased peak glucose concentration and reduced glucose clearance compared with WT mice (Fig. 1a). Increased area under curve (AUC) values indicated that $Irp2^{-/-}$ mice had developed glucose intolerance already at age of 2.5-months without worsening over time in mice at ages 5, 12–16, and 18 months (Fig. 1b). At all ages, $Irp2^{-/-}$ mice showed significantly higher elevated fasting glucose levels compared with WT mice (Fig. 1c). WT and $Irp2^{-/-}$ mice had indistinguishable body weights at 3.5-months of age; however, starting around 7-months of age, body weight of $Irp2^{-/-}$ mice was significantly lower compared with WT mice, and $Irp2^{-/-}$ mice showed reduced fat mass and increased lean mass (Supplementary Fig. 1a–c). $Irp2^{-/-}$ female mice (10- and 20-weeks of age) also displayed significant but less pronounced glucose intolerance compared with males (Supplementary Fig. 2a, b), and therefore, subsequent studies were performed solely in male mice.

To determine whether impaired glucose clearance in $Irp2^{-/-}$ mice is due to insulin resistance, insulin tolerance tests were performed in 7-month-old WT and $Irp2^{-/-}$ mice. Comparable glucose disposal rates suggested glucose intolerance in $Irp2^{-/-}$ mice was not a consequence of insulin resistance (Fig. 1d). We next performed euglycemic-hyperinsulinemic clamps in overnight fasted 7-month-old mice in order to analyze whole-body and organ-specific insulin action in more detail. When exogenous insulin was administered to $Irp2^{-/-}$ mice, they displayed increased insulin sensitivity evident by higher glucose infusion rates compared with age- and body composition-matched WT mice (Fig. 1e). Comparable whole-body glucose turnover rates (Fig. 1f) in both genotypes suggested improved insulin sensitivity in $Irp2^{-/-}$ mice is not caused by peripheral alterations in insulin action, but rather a more pronounced suppression of hepatic glucose production by insulin (Fig. 1g). Taken together, these data suggest that glucose intolerance in $Irp2^{-/-}$ mice is primarily related to impaired β-cell function.

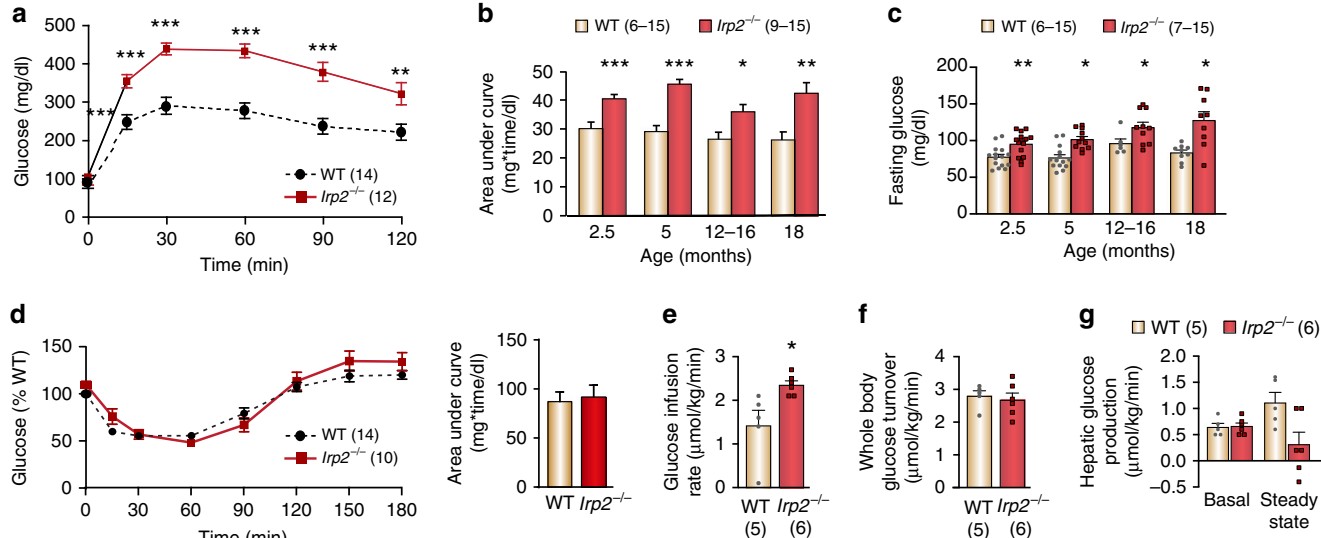

**Fig. 1 Glucose intolerance in *Irp2*$^{-/-}$ mice is not caused by impaired peripheral and hepatic insulin action. a** Intraperitoneal glucose tolerance tests (ipGTTs) for 5-month-old WT and *Irp2*$^{-/-}$ mice (2 g glucose/kg body weight). **b** Glucose AUC calculated from ipGTTs for WT and *Irp2*$^{-/-}$ mice at the indicated ages. **c** Plasma glucose concentrations for fasted WT and *Irp2*$^{-/-}$ mice. **d** Insulin tolerance tests (ITTs) performed in random-fed 5-month-old WT and *Irp2*$^{-/-}$ mice. Glucose AUC graph is shown on the right. **e–g** Euglycemic-hyperinsulinemic clamp experiments conducted in fasted 7-month-old WT and *Irp2*$^{-/-}$ mice. Glucose infusion rate (**e**), whole-body glucose turnover rate (**f**), and hepatic glucose production (**g**). Data in **a–g** are expressed as means ± s.e.m., unpaired two-tailed Student's *t* test, \*$p < 0.05$, \*\*$p < 0.01$, \*\*\*$p < 0.001$ relative to WT mice. The number of mice is indicated in parentheses. Source data are provided as a Source Data file.

**Insulin secretion is blunted in *Irp2*$^{-/-}$ mice.** To determine whether diabetes in *Irp2*$^{-/-}$ mice is caused by insulin insufficiency, we measured plasma insulin levels after intraperitoneal glucose injection. In the fasted state (0 min), basal insulin levels in 7- and 18-month-old *Irp2*$^{-/-}$ mice were similar to their WT controls (Fig. 2a, b). Intraperitoneal glucose injection in 7-month-old *Irp2*$^{-/-}$ mice is followed by an increase in plasma insulin concentrations from baseline levels, but is blunted compared with WT mice, and was weaker in 18-month-old mice, suggesting an age-dependent effect (Fig. 2a, b). To measure pancreatic β-cell sensitivity in response to elevations in plasma glucose, hyperglycemic clamps were carried out in overnight fasted 7-month-old WT and *Irp2*$^{-/-}$ mice. *Irp2*$^{-/-}$ mice showed fasting hyperglycemia (WT, 6.06 ± 0.23 versus *Irp2*$^{-/-}$, 7.17 ± 0.39 mmol glucose/L; $p = 0.02$; Fig. 2c) and reduced basal plasma insulin concentrations (WT, 1.87 ± 0.773 versus 0.489 ± 0.105 ng/ml insulin, $p < 0.05$; Fig. 2d). Insulin secretion in response to hyperglycemia (~18 mM) is blunted in *Irp2*$^{-/-}$ mice (Fig. 2d). Calculation of the AUC during the first phase (0–15 min) and steady-state second phase (60–105 min) of the clamp showed that insulin secretion was reduced by 62% ($p < 0.05$) and 67% ($p < 0.05$), respectively, in *Irp2*$^{-/-}$ mice compared with WT mice (Fig. 2e). These results indicate that glucose intolerance in *Irp2*$^{-/-}$ mice is caused by impaired insulin secretion from β cells.

**Irp2 deficiency causes proinsulin accumulation in β cells.** To determine the basis for reduced insulin secretion in *Irp2*$^{-/-}$ mice, pancreatic insulin content was quantified in WT and *Irp2*$^{-/-}$ mice. The total pancreatic insulin was reduced in 2.5-, 7.5-, and 18-month-old *Irp2*$^{-/-}$ mice compared with age- and weight-matched WT mice (Fig. 3a). By contrast, pancreatic proinsulin content and the proinsulin-to-insulin (P/I) ratio significantly increased in 2.5-, 7.5-, and 18-month-old *Irp2*$^{-/-}$ mice compared with WT mice (Fig. 3b, c). The plasma P/I ratio also significantly increased in 7.5-month-old *Irp2*$^{-/-}$ mice compared with WT mice (Fig. 3d). Reduced insulin content in *Irp2*$^{-/-}$ mice was not due to changes in insulin transcription as transcript levels of the

two insulin genes, *Ins1* and *Ins2*, were similar to WT islets (Fig. 3e). Morphometric quantification of insulin-stained paraffin-embedded pancreatic sections showed that islet area and β-cell mass in 2.5- and 7.5-month-old *Irp2*$^{-/-}$ mice were similar to age-matched WT mice, although islet area and β-cell mass tended to be reduced in 18-month-old *Irp2*$^{-/-}$ mice (Fig. 3f, g; Supplementary Fig. 3a–d). Insulin, glucagon, and the glucose transporter Glut2 immunostaining displayed normal subcellular localization, showing islet morphology is normal in *Irp2*$^{-/-}$ islets (Supplementary Fig. 4a, b). These results suggest that proinsulin processing to mature insulin is impaired in *Irp2*$^{-/-}$ β cells.

We next assessed the secretory responses of WT and *Irp2*$^{-/-}$ islets under conditions of basal (2.5 mM) glucose and high (16.7 mM) glucose concentrations in a 1-h static assay. Consistent with pancreatic studies, insulin content decreased, and proinsulin content and the P/I ratio increased in *Irp2*$^{-/-}$ islets compared with WT islets (Fig. 3h–j). Insulin secretion was blunted in *Irp2*$^{-/-}$ islets under high glucose compared with WT islets (Fig. 3k), while proinsulin secretion increased in *Irp2*$^{-/-}$ islets under basal glucose compared with WT islets and further increased under high glucose (Fig. 3l). To account for reduced insulin content in *Irp2*$^{-/-}$ islets, insulin secretion was calculated as a percentage of total insulin content. Normalization of insulin secretion to total islet insulin content showed that *Irp2*$^{-/-}$ islets secreted similar amount of insulin compared with WT, suggesting that impaired insulin secretion results mostly from reduced insulin content (Fig. 3m). Normalization of proinsulin secretion to total proinsulin content showed that *Irp2*$^{-/-}$ islets secreted more of their total proinsulin compared with WT islets under high glucose, suggesting that proinsulin secretion is enhanced by Irp2 deficiency (Fig. 3n). Together, these data suggest that impaired glucose-stimulated insulin secretion in *Irp2*$^{-/-}$ islets is primarily caused by reduced insulin content, and that this defect is intrinsic to the β cell.

**Cellular iron homeostasis is dysregulated in *Irp2*$^{-/-}$ islets.** Given that Irp2 controls cellular iron homeostasis by regulating

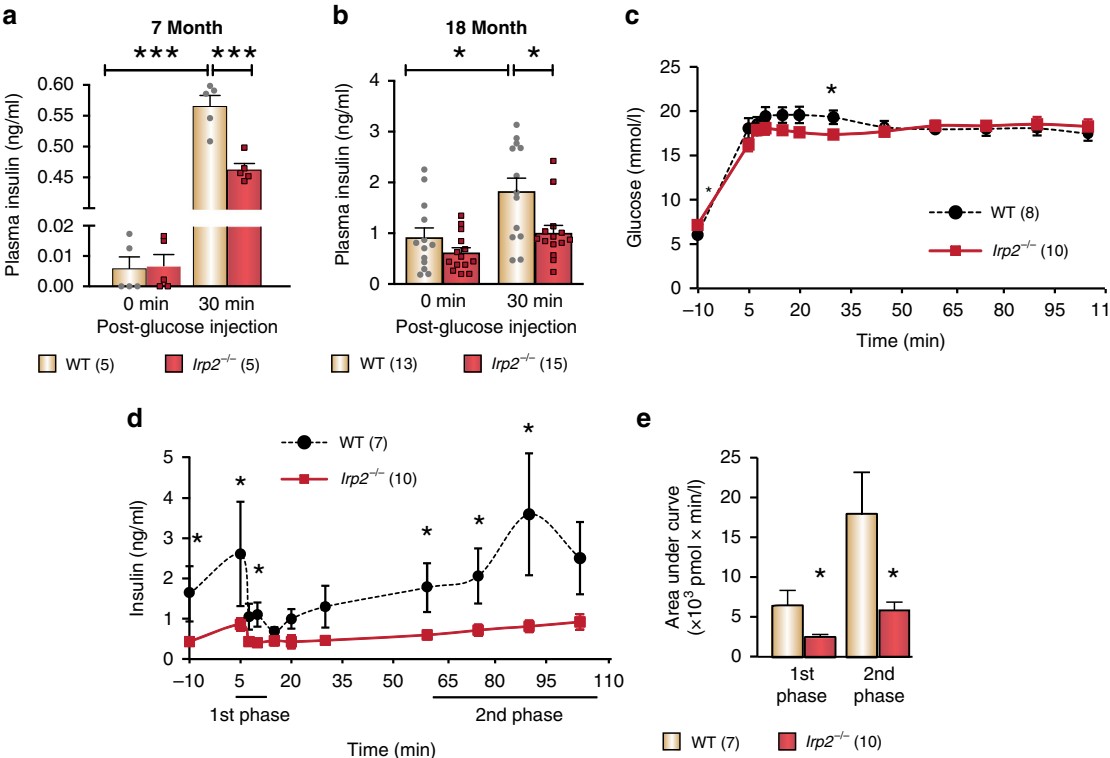

**Fig. 2 Glucose-stimulated insulin secretion is blunted in Irp2$^{-/-}$ mice. a, b** Plasma insulin levels in fasted 7-month-old (**a**) and 18-month-old (**b**) WT and Irp2$^{-/-}$ mice at (0 min) or after glucose injection (30 min). Data are expressed as means ± s.e.m., one-way ANOVA with Tukey's multiple comparison test, *$p < 0.05$, ***$p < 0.001$. **c–e** Hyperglycemic clamp performed in 7-month-old fasted WT and Irp2$^{-/-}$ mice. Glucose (**c**) and insulin (**d**) levels measured at the indicated times. **e** Insulin AUC corresponding to the first phase (0–15 min) and second phase (60–105 min) of insulin secretion are shown. Data in **c–e** are expressed as means ± s.e.m., unpaired two-tailed Student's t test, *$p < 0.05$ relative to WT mice. The number of mice is indicated in parentheses. Source data are provided as a Source Data file.

the expression of ferritin and TfR1, the levels of Fth1- and Ftl1-subunits, and TfR1 were assessed in WT and Irp2$^{-/-}$ islets. Western blot analysis showed no detectable Irp2 and no change in Irp1 levels in Irp2$^{-/-}$ islets (Fig. 4a). Ftl1 levels were similar in WT and Irp2$^{-/-}$ islets, while Fth1 was not detected in WT islets, but notably increased in Irp2$^{-/-}$ islets (Fig. 4a). As expected, TfR1 protein and mRNA levels were lower in Irp2$^{-/-}$ islets compared with WT islets, consistent with destabilization of TfR1 mRNA caused by Irp2 loss (Fig. 4a, b). Ftl1 mRNA levels were similar in WT and Irp2$^{-/-}$ islets, but unexpectedly, Fth1 mRNA levels decreased in Irp2$^{-/-}$ islets, which may be a transcriptional response to compensate for increased Fth1 protein (Fig. 4b). Double-immunofluorescence studies revealed prominent ferritin costaining with insulin, and reduced TfR1 costaining with insulin in Irp2$^{-/-}$ β cells compared with WT in agreement with western blot analysis (Fig. 4c, d). Consistent with reduced TfR1 levels, total iron content, as quantified by inductively coupled proton optical emission spectroscopy (ICP-OES), was reduced by 48% in Irp2$^{-/-}$ islets compared with WT islets, whereas the content of other metals (Cu, Mn, and Zn) was unchanged (Table 1). These data show that Irp2 deficiency in β cells reduces TfR1 levels and iron content, and increases ferritin levels and iron sequestration, causing functional cellular iron deficiency.

**Iron normalizes insulin secretion in Irp2$^{-/-}$ mice.** Because cellular iron homeostasis is dysregulated in Irp2$^{-/-}$ islets, we questioned whether insulin and proinsulin secretion can be normalized by iron supplementation in Irp2$^{-/-}$ islets. For these experiments, insulin and proinsulin secretion were first assessed in freshly isolated WT and Irp2$^{-/-}$ islets under basal glucose

(2.5 mM). Islets were then cultured overnight in medium containing the iron chelator desferrioxamine (DFO) or iron (ferric ammonium citrate (FAC)) and assessed for insulin and proinsulin secretion under high glucose (16.7 mM). Consistent with islet studies in Fig. 3k, l, insulin secretion was blunted and proinsulin secretion increased in Irp2$^{-/-}$ islets compared with WT islets (Fig. 5a, b). DFO reduced insulin secretion and increased proinsulin secretion in WT islets compared with untreated WT islets (Fig. 5a, b), while FAC had no significant effect on insulin and proinsulin secretion in WT islets, but normalized insulin and proinsulin secretion in Irp2$^{-/-}$ islets to levels observed in untreated WT islets (Fig. 5a, b). Likewise, DFO reduced insulin content and increased proinsulin content in WT islets, while FAC normalized insulin content and partially normalized proinsulin content in Irp2$^{-/-}$ islets to levels observed in untreated WT islets (Fig. 5c, d). FAC also restored the proinsulin-to-insulin ratio in Irp2$^{-/-}$ islets to the untreated WT ratio (Fig. 5e). These results show that iron normalizes proinsulin and insulin secretion and content in Irp2$^{-/-}$ islets.

To determine whether iron supplementation normalizes insulin and proinsulin content in vivo, WT and Irp2$^{-/-}$ mice were intraperitoneally injected with iron dextran or PBS as a control for 5 days, after which mice were sacrificed, and pancreatic insulin and proinsulin content were quantified. Iron dextran increased pancreatic and liver iron content in WT and Irp2$^{-/-}$ mice compared with untreated mice (Fig. 5f, g). Iron dextran partially normalized insulin content and fully normalized proinsulin content in Irp2$^{-/-}$ mice to WT levels (Fig. 5h, i). The partial normalization of insulin content in Irp2$^{-/-}$ mice by iron may be due to reduced TfR1 expression in these islets (Fig. 4a). Of note, iron dextran reduced insulin content in WT

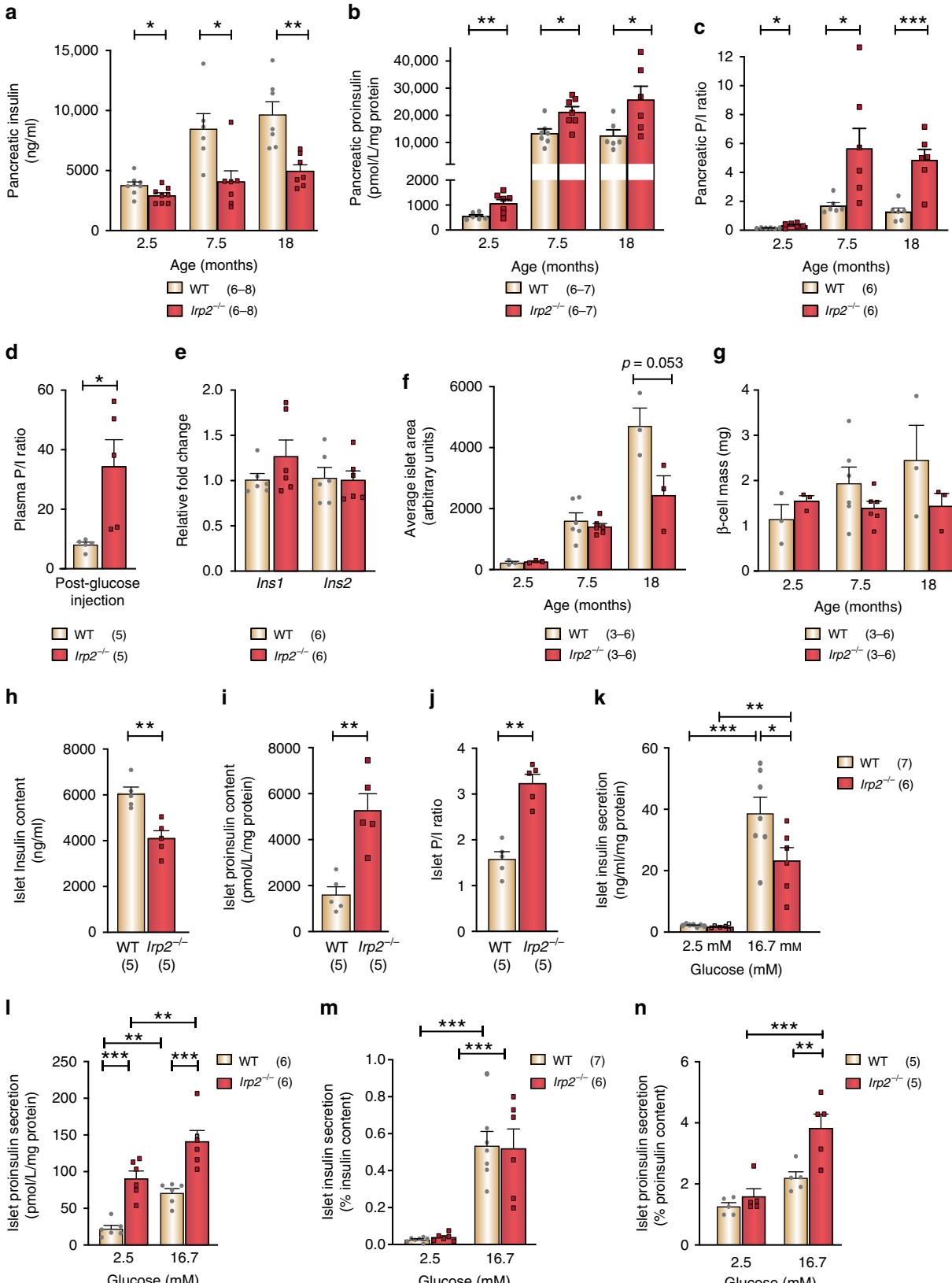

mice (Fig. 5h), suggesting that pancreatic iron overload may impair insulin production in normal mice. Collectively, these data indicate that maintenance of cellular iron homeostasis by Irp2 is critical for normal proinsulin processing and insulin secretion.

**Impaired insulin secretion in Irp2-depleted insulinoma cells**. The role of Irp2 in insulin production and secretion was further studied in rat insulinoma INS-1 832/13 cells depleted of Irp2 by stable expression of a short-hairpin Irp2 RNA (shIrp2 RNA). These cells provided a tool for elucidating the mechanism

**Fig. 3 Irp2 deficiency leads to proinsulin accumulation in β cells. a** Quantification of pancreatic insulin content, **b** proinsulin content, and **c** pancreatic proinsulin-to-insulin ratio (P/I) in 2.5-, 7.5-, and 18-month-old WT and $Irp2^{-/-}$ mice. **d** Plasma P/I ratio in 7.5-month-old WT and $Irp2^{-/-}$ mice. **e** qPCR shows no difference in $Ins1$ and $Ins2$ expression in WT and $Irp2^{-/-}$ islets from 10-month-old mice. Values are normalized to β-actin mRNA and are expressed as fold change relative to WT. **f, g** Quantification of islet area (**f**) and β-cell mass (**g**) in insulin-stained paraffin-embedded pancreatic sections from 2.5-, 7.5-, and 18-month-old WT and $Irp2^{-/-}$ mice. Mass was calculated by multiplying the fraction of insulin-positive β-cell area by pancreatic wet weight. **h** The total islet insulin content, **i** proinsulin content, and **j** islet P/I ratio measured in WT and $Irp2^{-/-}$ islets from 7.5-month-old mice. **k, l** Glucose-stimulated insulin (**k**) and proinsulin secretion (**l**) measured in islets under basal (2.5 mM) glucose and after stimulation with high (16.7 mM) glucose for 1 h and normalized to total islet protein. **m, n** Insulin (**m**) and proinsulin (**n**) secretion measured in islets in (**k, l**) normalized to total islet insulin or proinsulin content. Data are expressed as means ± s.e.m., unpaired two-tailed Student's $t$ test for **a–j** and a one-way ANOVA with Tukey's multiple comparisons test for **k–n**, *$p < 0.05$, **$p < 0.01$, ***$p < 0.001$. The number of mice is indicated in parentheses. Source data are provided as a Source Data file.

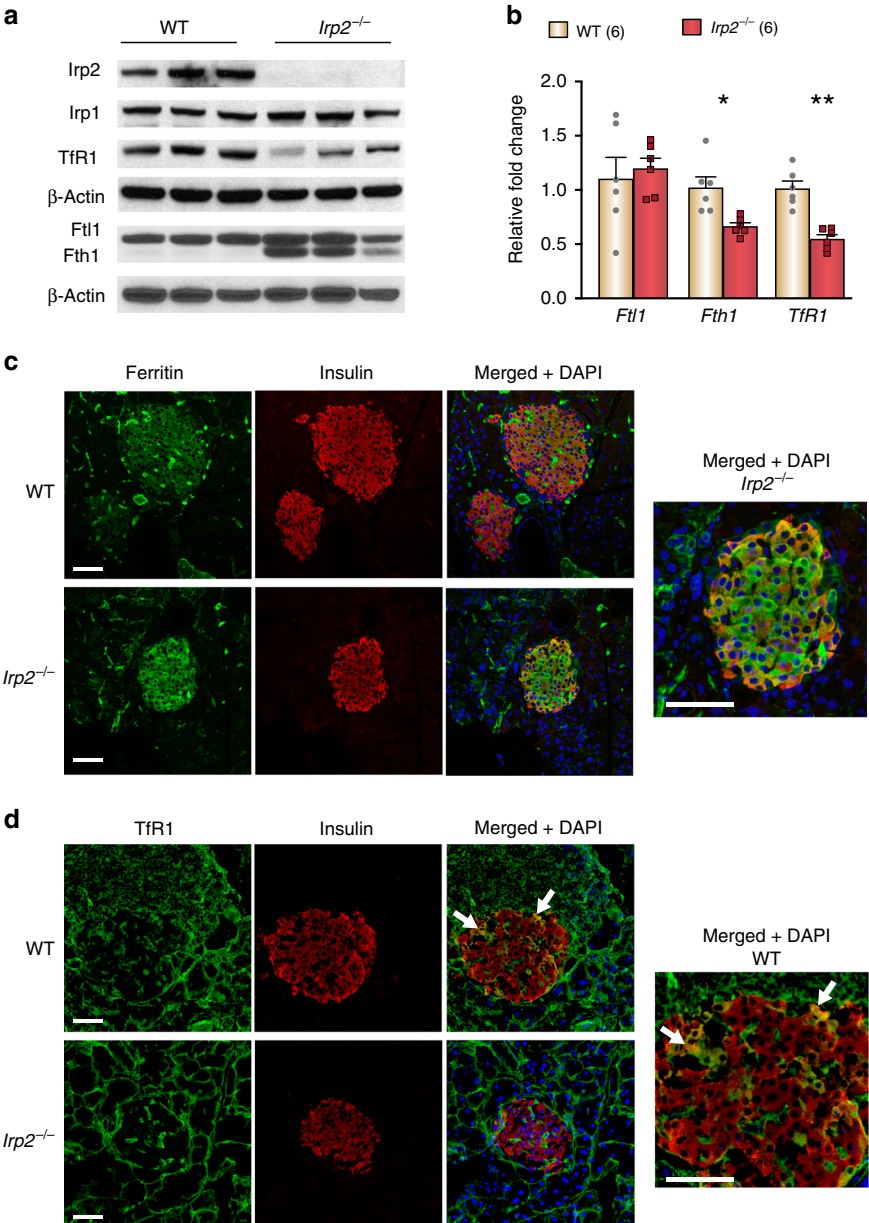

**Fig. 4 Cellular iron homeostasis is dysregulated in $Irp2^{-/-}$ islets. a** Western blot analysis of Irp2, Irp1, TfR1, ferritin (Fth1 and Ftl1) in WT, and $Irp2^{-/-}$ islets from 8-month-old mice ($n = 3$ mice per genotype). β-Actin is a loading control. Note that murine Fth1 migrates slightly faster than Ftl1 on SDS-PAGE[67]. **b** qPCR of $Ftl1$, $Fth1$, and $TfR1$ in WT and $Irp2^{-/-}$ islets from 8-month-old mice. Values are normalized to β-actin mRNA and expressed as relative fold change to WT islets ($n = 6$ mice per genotype). Data are expressed as means ± s.e.m., unpaired two-tailed Student's $t$ test, *$p < 0.05$, **$p < 0.01$. **c, d** Representative paraffin-embedded pancreatic sections from 8-month-old WT and $Irp2^{-/-}$ mice immunostained with antibodies to ferritin (green) and insulin (red) (**c**) or antibodies to TfR1 (green) and insulin (red) (**d**). The insulin antibody detects both insulin and proinsulin. Nuclei are stained with DAPI (blue). Scale bar: 50 μm. Source data provided as a Source Data file.

**Table 1 Metal content in WT and Irp2⁻/⁻ islets.**

|    | WT (7) | Irp2⁻/⁻ (6) |
|----|--------|-------------|
| Cu | 53.0 ± 8.8 | 48.8 ± 10.3 |
| Fe | 475.4 ± 48.1 | 250.0 ± 46.7** |
| Mn | 10.4 ± 1.23 | 10.2 ± 3.8 |
| Zn | 779.9 ± 41.1 | 640.0 ± 61.3 |

Data are expressed as means (pg metal/islet) ± s.e.m., compared by unpaired two-tailed Students $t$ test, **$p < 0.001$ relative to WT. Metal content was determined by ICP-OES. The number of mice use are indicated in parentheses. Age of mice, 10-month-old males. Source data are provided as a Source Data File

underlying β-cell dysfunction caused by Irp2 deficiency. Western blot analysis confirmed reduced Irp2 expression in shIrp2 cells concomitant with the expected increase in ferritin and reduction in TfR1 levels compared with cells expressing empty vector (EV) (Fig. 6a). Treatment of cells with DFO stabilized Irp2, reduced ferritin and increased TfR1 levels, whereas FAC reduced Irp2, increased ferritin and reduced TfR1 levels in both EV and shIrp2 cells (Fig. 6a). Irp2 RNA-binding activity, as measured by an RNA-electrophoretic mobility shift assay (RNA-EMSA), decreased in shIrp2 cells concomitant with increased Irp1 RNA-binding activity, consistent with reduced total iron content in these cells (Fig. 6b, c). Consistent with Irp2⁻/⁻ islets studies, shIrp2 cells displayed reduced insulin content, increased proinsulin content, as well as reduced glucose-stimulated insulin secretion and increased proinsulin secretion compared with EV cells (Fig. 6d–g). When the amount of secreted insulin was normalized to insulin content, shIrp2 cells secreted slightly less insulin compared with EV cells, suggesting that secretion may be somewhat impaired (Fig. 6h). No significant difference in proinsulin secretion was observed under high glucose in EV and shIrp2 cells after normalization to total proinsulin content (Fig. 6i). FAC supplementation restored insulin and proinsulin content, and glucose-stimulated insulin and proinsulin secretion to levels observed in EV cells (Fig. 6d–g). Together, these results show that shIrp2 cells recapitulate the abnormal proinsulin–insulin phenotype observed in primary Irp2⁻/⁻ islets.

**Irp2 deficiency impairs Fe–S cluster protein function.** Previous studies have shown that the activities of Fe–S cluster containing respiratory complexes I, II, and III, and aconitase are reduced in motor neurons of Irp2⁻/⁻ mice[18] and in hepatocytes of Irp1⁻/⁻; Irp2⁻/⁻ mice[25]. We therefore examined Fe–S cluster biosynthesis in Irp2-deficient INS-1 832/13 cells by measuring the activity of mitochondrial (m)- and cytosolic (c)-aconitases, and complex I, as well as complex IV, which lacks Fe–S clusters. For these studies, additional Irp2-deficient INS-1 832/13 cells lines were generated using CRISPR/Cas9-editing (sgIrp2.1 and sgIrp2.2), and Irp2 depletion and iron regulation were validated in these cell lines by western blot analysis (Fig. 7a). The total iron content was reduced in sgIrp2.1 and sgIrp2.2 cells compared with control parental INS-1 cells (Fig. 7b). M-aconitase and c-aconitase activity, and complex I activity were lower in sgIrp2.1 and sgIrp2.2 cells, and shIrp2 cells, respectively, compared with control cells, and FAC normalized activities of all proteins to levels observed in control cells (Fig. 7c–e). Reduced c-aconitase activity in sgIrp2.1 and sgIrp2.2 cells is consistent with increased Irp1 RNA-binding activity observed in shIrp2 cells (Figs. 7b, 6b). Complex IV activity was not significantly altered in shIrp2 cells (Fig. 7f). ATP production, as measured under basal glucose (5 mM) and after stimulation with 15 mM glucose, was blunted in sgIrp2.1 and sgIrp2.2 cells compared with control cells with FAC partially normalizing ATP production in sgIrp2.1 and sgIrp2.2

cells (Fig. 7g). Together, these data suggest that cellular iron deficiency caused by loss of Irp2 impairs Fe–S cluster biosynthesis, resulting in reduced mitochondrial and cytosolic Fe–S protein function and ATP production.

**The UPR is activated in Irp2-depleted insulinoma cells.** Accumulation of proinsulin in the endoplasmic reticulum (ER) lumen can lead to its misfolding or unfolding triggering ER stress and activating the unfolded protein response (UPR)[26]. Increased proinsulin levels in Irp2-deficient INS-1 cells suggested that the UPR might be activated in these cells. UPR activation was assessed using the UPR indicator markers, phosphorylated eukaryotic translation initiation factor 2α (eIF2α-P), a protein that attenuates translation, and Grp78/BiP, an ER chaperone that increases protein folding[27,28]. For these experiments, eIF2α-P and Grp78/BiP levels were examined in control, sgIrp2.1, and sgIrp2.2 cells under basal glucose (5 mM) and after stimulation with 15 mM glucose to induce stress. eIF2α-P increased in sgIrp2.1 and sgIrp2.2 cells under basal and high glucose compared with control cells, while eIF2α-total levels were unchanged (Fig. 7h). Grp78/BiP was highly expressed in all cells and under all glucose conditions at similar levels (Fig. 7h). These data suggest eIF2α-P upregulation in Irp2-deficient INS-1 cells may provide a mechanism to attenuate proinsulin translation, ensuring cellular adaptation to iron deficiency.

**Loss of Irp2 impairs proinsulin translation.** The reduction in Fe–S cluster biosynthesis in Irp2-deficient INS-1 cells suggested the possibility that a Fe–S cluster protein related to proinsulin function might be impaired. After reviewing the literature, we focused on the radical SAM enzyme Cdkal1 that uses 4Fe–4S clusters to catalyze the methylthiolation of t⁶A to ms²t⁶A on adenosine 37 in tRNA$^{Lys}_{UUU}$[19,20] (Fig. 8a). This modification is critical for the accurate reading of lysine AAA and AAG codons in proinsulin, and its loss in β cells in Cdkal1 KO mice and in Cdkal1-deficient INS-1 cells results in reduced insulin content and secretion[20,29]. Because many Fe–S proteins depend on their Fe–S clusters for stability[30], we examined Cdkal1 levels in sgIrp2.1 and sgIrp2.2 cells, and in Irp2⁻/⁻ islets. Cdkal1 levels were modestly reduced in sgIrp2.1 and sgIrp2.2 cells, but were noticeably reduced with DFO compared with control cells (Fig. 8b). Notably, Cdkal1 levels were reduced in Irp2⁻/⁻ islets compared with WT islets (Fig. 8c). Brambillasca et al.[29] reported that (pro) chromogranin A (CgA), a protein that is co-sorted and co-processed with proinsulin in insulin secretory granules[31], was reduced in Cdkal1-deficient INS-1 cells. Similarly, we found reduced CgA levels in Irp2⁻/⁻ islets compared with WT islets (Fig. 8c). Pcsk1, a prohormone convertase that processes proinsulin and CgA to their mature forms, was unchanged in Irp2⁻/⁻ islets (Fig. 8c). These data show that Irp2 deficiency results in reduced levels of Cdkal1 as well as CgA in Irp2⁻/⁻ islets.

Reduced Cdkal1 expression in Irp2⁻/⁻ islets suggested that the ms²t⁶A modification may also be reduced. To test this premise, we used liquid chromatography-coupled mass spectrometry (LC-MS) to measure ms²t⁶A levels in control, sgIrp2.1, and sgIrp2.2 cells. Levels of ms²t⁶A were significantly reduced in sgIrp2.1 cells, and showed a tendency to be reduced in sgIrp2.2 cells compared with control INS-1 cells (Fig. 8d; Supplementary Fig. 5a). Iron supplementation restored ms²t⁶A levels in sgIrp2.1 and sgIrp2.2 cells to levels in control cells (Fig. 8d). No significant differences were observed in t⁶A levels in sgIrp2.1 and sgIrp2.2 cells compared with control cells (Fig. 8e; Supplementary Fig. 5b).

We next questioned the impact of reduced ms²t⁶A levels on proinsulin translation in Irp2-deficient INS-1 cells. Rat, mouse, and human proinsulin each contain a lysine residue at the

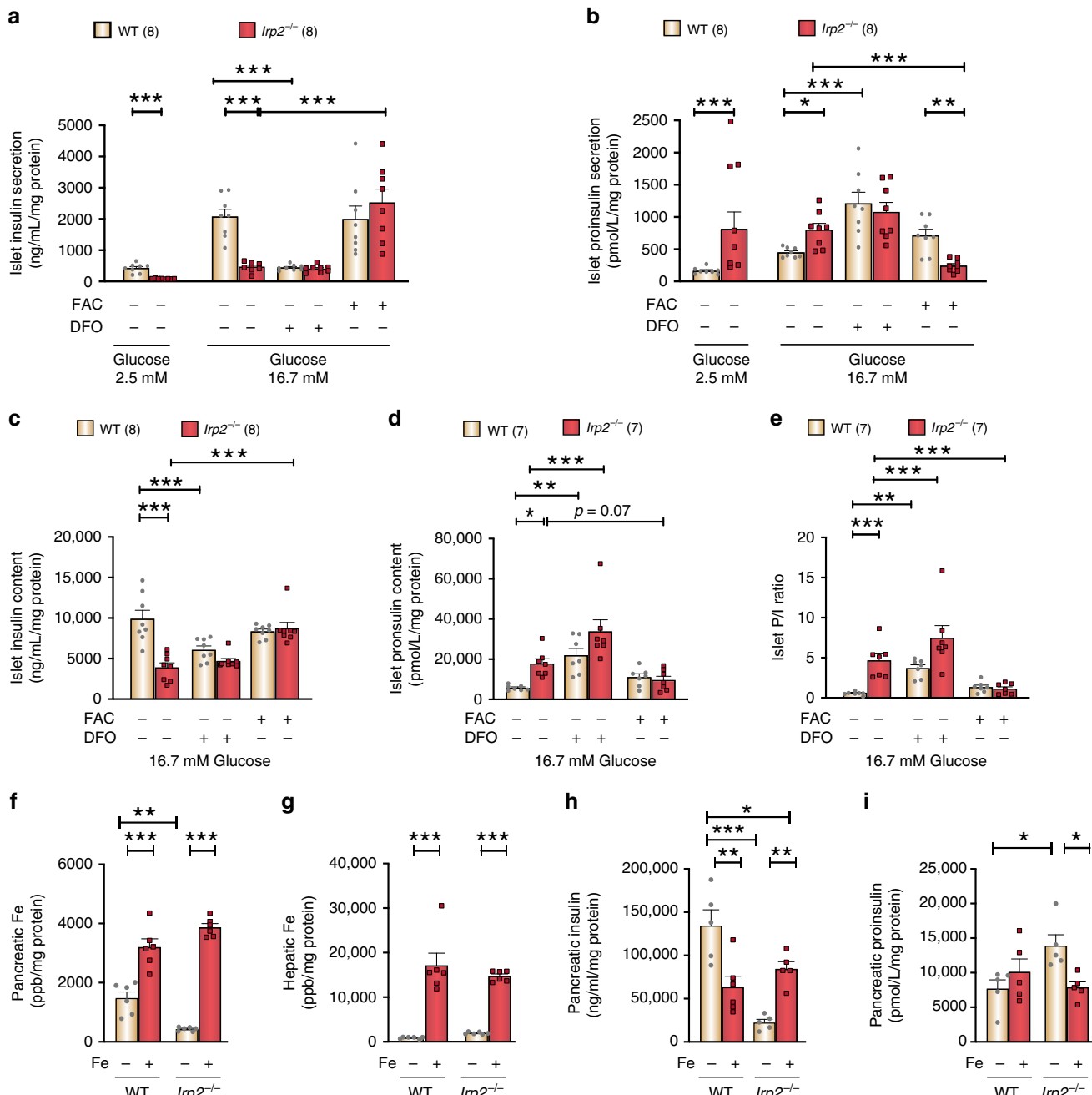

**Fig. 5 Iron normalizes insulin and proinsulin secretion and content in *Irp2*<sup>−/−</sup> islets and mice. a–e** Glucose-stimulated insulin (**a**) and proinsulin secretion (**b**), total insulin (**c**), and proinsulin (**d**) content and islet P/I ratio (**e**) in WT and *Irp2*<sup>−/−</sup> islets from 7.5-month-old mice. Insulin and proinsulin secretion were assessed under basal glucose (2.5 mM) and after culture overnight in the presence of DFO (50 μM) or FAC (50 μg/ml) under high glucose (16.7 mM). The number of mice used is indicated in parentheses. **f–i** WT and *Irp2*<sup>−/−</sup> mice were i.p.-injected with iron dextran or PBS alone for 5 days, and pancreatic (**f**) and hepatic (**g**) iron content was quantified by ICP-MS and normalized to total cellular protein. Pancreatic insulin (**h**) and proinsulin (**i**) content were normalized to total cellular protein. Data are expressed as means ± s.e.m., one-way ANOVA with Tukey's multiple comparisons test for **a–i**, *$p < 0.05$, **$p < 0.01$, ***$p < 0.001$. Source data are provided as a Source Data file.

Pcsk1 cleavage site between the A-chain and the C-peptide, and lysine residues within the B-chain, and misreading of these lysine codons could impair proinsulin cleavage and/or folding. To examine proinsulin translation accuracy, control, sgIrp2.1, and sgIrp2.2 INS-1 cells were grown with or without supplemental iron, metabolically labeled with ³H-leucine and ¹⁴C- lysine, and the relative incorporation of ¹⁴C-lysine versus ³H-leucine in immunoprecipitated proinsulin was determined. The relative incorporation of ¹⁴C-lysine versus ³H-leucine in

proinsulin in sgIrp2.1 and sgIrp2.2 cells was significantly reduced compared with control cells, and iron supplementation fully normalized proinsulin ¹⁴C-lysine incorporation in sgIrp2.1 cells and partially in sgIrp2.2 cells (Fig. 8f). Together, these data show that reduced Cdkal1 function caused by Irp2 deficiency reduces the ms²t⁶A modification in tRNA<sup>Lys</sup><sub>UUU</sub>, leading to misreading of lysine codons in proinsulin, which impairs proinsulin processing and insulin production (Fig. 8g).

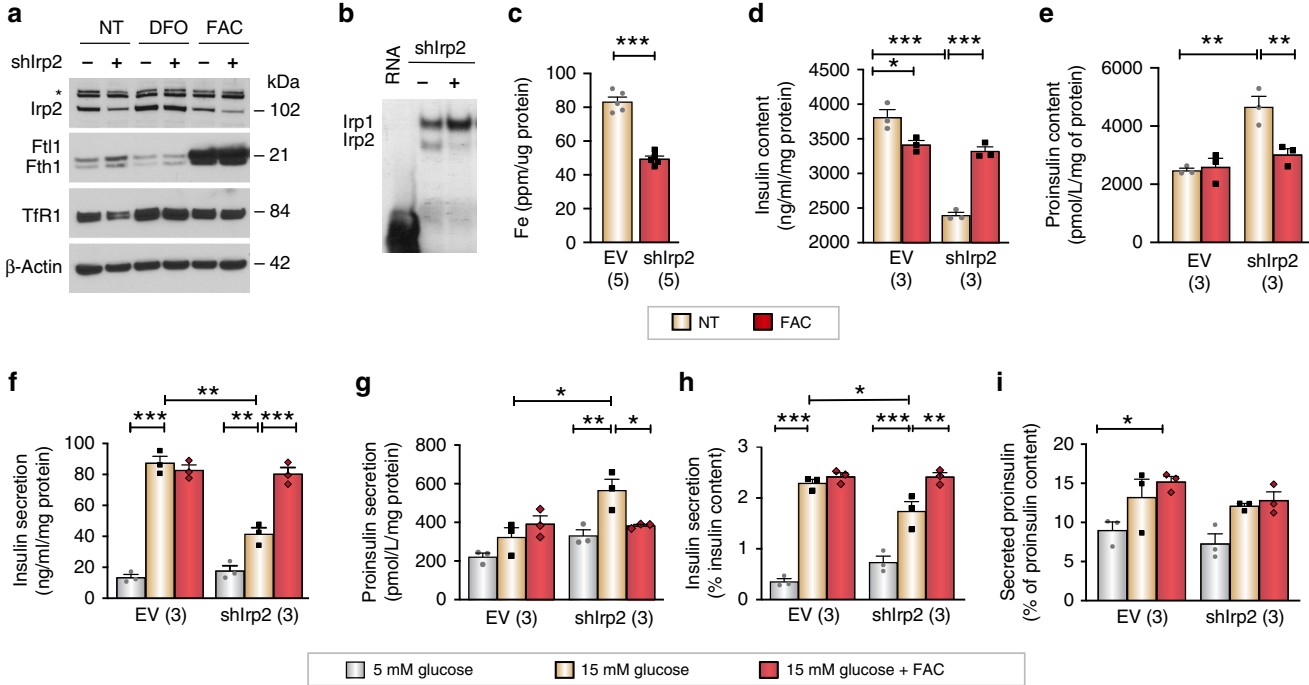

**Fig. 6 Iron normalizes insulin and proinsulin secretion in Irp2-deficient insulinoma cells. a** Western blotting of Irp2, Ftl1, Fth1, and TfR1 in INS-1 832/13 insulinoma cells expressing a shIrp2 or empty vector (EV) control (−). Cells were grown in medium with no treatment (NT) or in the presence of DFO (50 μM) or FAC (50 μg/mL) for 18 h. β-Actin is a loading control. Asterisk, nonspecific band. **b** Irp1 and Irp2 RNA-binding activity was assessed in EV (−) and shIrp2 (+) cell lysates by RNA-EMSA. Whole-cell lysates were incubated with a [32]P-labeled ferritin IRE RNA probe followed by resolution of the Irp1- and Irp2-RNA complexes on non-denaturing polyacrylamide gels. Irp1- and Irp2-RNA complexes and [32]P-labeled ferritin IRE are indicated. **c** The total iron content in EV and shIrp2 cells as quantified by ICP-MS and normalized to total cellular protein. **d–g** The total insulin (**d**) and proinsulin content (**e**) and glucose-stimulated insulin (**f**) and proinsulin (**g**) secretion in EV and shIrp2 cells grown in the presence of FAC overnight and assayed under basal (5 mM) glucose and after stimulation with high (15 mM) glucose for 1 h. **h, i** Samples in **f** and **g** were normalized to total insulin content (**f**) or total proinsulin (**g**) content. $n = 3$ independent biological experiments. Data are expressed as means ± s.e.m., one-way ANOVA with Tukey's multiple comparisons test for **d–i** and unpaired two-tailed Student's $t$ test for **c**, **$p < 0.01$. Source data are provided as a Source Data file.

## Discussion

Iron overload is a known risk factor in the development of T2D[1–3], but the physiological and cellular impact of iron deficiency in β-cell dysfunction and diabetes are not yet fully understood. In this report, we demonstrate that cellular iron deficiency as a consequence of Irp2 loss in β cells causes diabetes. We found that iron deficiency in β cells impairs Fe–S cluster biosynthesis that reduces Cdkal1 function and the ms[2]t[6]A modification in tRNA[Lys]$_{UUU}$. As a consequence, lysine codons in proinsulin are misread, proinsulin processing is impaired, and insulin content and secretion are reduced (Fig. 8g). Our work thus implicates iron deficiency as a potential mechanism for β-cell dysfunction in humans.

Single-nucleotide polymorphisms (SNPs) in the *CDKAL1* gene locus are a strong risk factor for the development of T2D[21–24] and are associated with impaired glucose metabolism and insulin secretion in humans[32–35]. The importance of Cdkal1 in β-cell function arises from studies in *Cdkal1* β-cell KO mice showing that the ms[2]t[6]A modification in tRNA[Lys]$_{UUU}$ is critical for the accurate reading of lysine AAA and AAG codons, notably during conditions that increase protein synthesis[20]. In this study, *Bacillus subtilis* wild-type and *ygeV* mutants (lack the ms[2]t[6]A modification) were transformed with dual-luciferase reporter gene containing replacement of a lysine codon critical for luciferase activity with either AAA or AAG. Upon IPTG induction, luciferase activity was reduced in *ygeV* mutants expressing either the AAA or AAG compared with wild-type, consistent with misreading of lysine codons caused by loss of vgeV[20]. In mammals, proinsulin synthesis constitutes ~30–50% of the total protein synthesis in the

β cell[36], and the ms[2]t[6]A modification in tRNA[Lys]$_{UUU}$ may be critical for efficient and accurate proinsulin translation. In agreement with studies in *Cdkal1* β-cell KO mice[20], proinsulin accumulation in Irp2-deficient INS-1 cells is associated with UPR activation of PERK-dependent eIF2α phosphorylation, which is known to reduce global translation[26–28]. Because β-cell mass was not reduced in 2.5- and 7.5-month-old *Irp2[−/−]* mice, this suggests that phosphorylated eIF2α-mediated translational attenuation may allow Irp2-deficient β cells to adapt to cellular iron deficiency, thereby providing protection against UPR activation of apoptotic signaling pathways.

Cdkal1 deficiency in INS-1 cells has been shown to be associated with reduced expression of proinsulin as well as other insulin secretory granule proteins, such as (pro) CgA and (pro) ICA512/IA-2, that are processed with proinsulin in insulin secretory granules[29]. Consistent with this study, CgA expression was reduced in *Irp2[−/−]* islets compared with WT islets. It is therefore possible that Cdkal1 deficiency may affect the function of other insulin secretory granule proteins that could also contribute to insulin secretory defect observed in *Irp2[−/−]* mice.

Irp2 deficiency in β cells is associated with impaired mitochondrial function characterized by reduced complex I and m-aconitase activities, and reduced ATP production. Reduced activity of these proteins is likely due to impaired Fe–S cluster biosynthesis caused by iron deficiency in Irp2-deficient β cells. ATP is required for exocytosis of insulin secretory granules[37], and reduced ATP production in *Irp2[−/−]* β cells could affect glucose-stimulated insulin secretion. Our data, however, showed no major difference in insulin secretion in *Irp2[−/−]* islets or Irp2-deficient

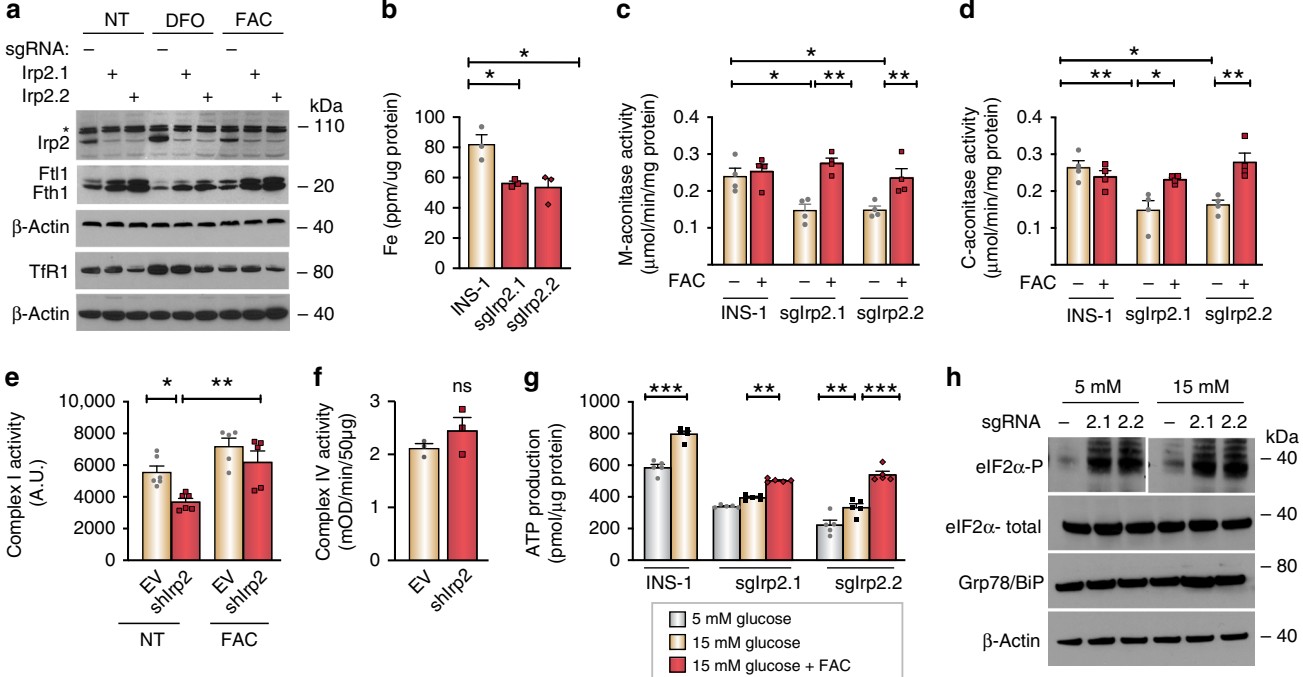

**Fig. 7 Irp2 deficiency impairs Fe–S cluster protein function and causes ER stress. a** CRISPR/Cas9 depletion of Irp2 in INS-1 832/13 cells (sgIrp2.1, sgIrp2.2) and control parental cells grown in the presence of FAC (10 μg/ml), DFO (50 μM), or no treatment (NT) for 18 h. Irp2, Fth1, Ftl1 and TfR1 were assessed by western blot analysis to show efficacy of Irp2 knockdown and appropriate iron regulation. β-Actin is a loading control. Asterisk, nonspecific band. **b** The total iron content measured by ICP-MS and normalized to total cellular protein ($n = 3$ independent biological experiments). **c, d** Mitochondrial (**c**) and cytosolic (**d**) aconitase activity in control, sgIrp2.1, and sgIrp2.2 INS-1 cells grown in medium with or without supplemental FAC and normalized to total cellular protein. **e, f** Complex I activity (**e**) and complex IV activity (**f**) in lysates from control EV and shIrp2 cells grown in medium with or without supplemental FAC and normalized to total cellular protein ($n \geq 3$ independent biological experiments). **g** ATP production in control INS-1, sgIrp2.1, and sgIrp2.2 cells grown in medium with or without supplemental FAC and assayed under basal (5 mM) glucose and after stimulation with 15 mM glucose for 1 h. ATP production was normalized to total cellular protein ($n = 5$ independent biological experiments). **h** Western blot analysis of eIF2α-P, eIF2α-total, and Grp78/BiP levels in sgIrp2.1 and sgIrp2.2 cells under basal glucose (5 mM) and after stimulation with 15 mM glucose. β-Actin is a loading control. Data are expressed as means ± s.e.m., unpaired two-tailed Student's $t$ test for **b** and **f** and one-way ANOVA with Tukey's multiple comparisons test for **c–e** and **g**, *$p < 0.05$; **$p < 0.01$, ***$p < 0.001$. Source data are provided as a Source Data file.

INS-1 832/13 cells compared with controls when insulin secretion is normalized to insulin content, suggesting that reduced insulin secretion in $Irp2^{-/-}$ mice and Irp2-deficient cells is mostly caused by reduced insulin content. Of note, studies have reported reduced glucose-stimulated ATP production and first-phase insulin secretion in $Cdkal1^{-/-}$ mice and in $Cdkal1$ β-cell KO mice, but how Cdkal1 deficiency impairs mitochondria function is not yet fully understood[20,38].

Our studies using global $Irp2^{-/-}$ mice reveal Irp2 as a critical regulator of β cell iron homeostasis. We show that the abnormal proinsulin phenotype in $Irp2^{-/-}$ mice is recapitulated in $Irp2^{-/-}$ islets and in Irp2-deficient INS-1 cells, indicating that this defect is intrinsic to the β cell. It is possible, however, that signals from other tissues may modulate β-cell function in $Irp2^{-/-}$ mice. Iron has been shown to negatively regulate adiponectin and leptin expression in adipocytes[39,40], and these adipokines are known to facilitate β-cell proliferation and survival[41–43]. In addition, Cdkal1 deficiency has been shown to regulate adipocyte differentiation in murine 3T3-L1 cells[44] and mitochondrial function in adipocyte-specific $Cdkal1$ KO mice[45] as well as regulation of growth hormone expression in pituitary adenomas[46], suggesting that reduced Cdkal1 function in cell types other than β cells might modulate β-cell function in $Irp2^{-/-}$ mice. The generation of β-cell-specific $Irp2$ KO mice will be useful to determine whether changes in iron content in other tissues affect β-cell function.

A recent study using a distinct $Irp2^{-/-}$ mouse strain reported that 18-month-old male $Irp2^{-/-}$ mice displayed hyperglycemia, insulin resistance as determined by ITTs, and no change in early-phase glucose-stimulated insulin secretion[47], while we found that 18-month-old male $Irp2^{-/-}$ mice displayed reduced pancreatic insulin content and glucose-stimulated insulin secretion compared with WT mice. An explanation for this discrepancy is not clear, but may stem from differences in strain backgrounds or different assay conditions.

Iron deficiency and its role in anemia is well known, but the physiological and cellular effects of iron deficiency on non-erythropoietic cells and tissues are less clear. There is one case report of a patient with iron deficiency anemia and diabetes that was attributed to autoreactive antibodies against the TfR1[48,49], suggesting the possibility that diabetes in this patient may be related to impaired proinsulin processing shown in our current study. Other studies in humans have shown that iron deficiency is associated with pulmonary arterial hypertension[50–52], and cardiovascular[53] and neurodegenerative diseases[54]. Studies in rats have revealed that chronic dietary iron deficiency causes mitochondrial dysfunction and remodeling of the pulmonary vasculature[55]. Similarly, TfR1 deficiency in murine dopaminergic neurons causes mitochondrial dysfunction and neuronal degeneration[56]. Recently, two patients with mutations in $IREB2$ have been identified that exhibit early onset and progressive neurological disease, and microcytic anemia[57,58]; however, insulin

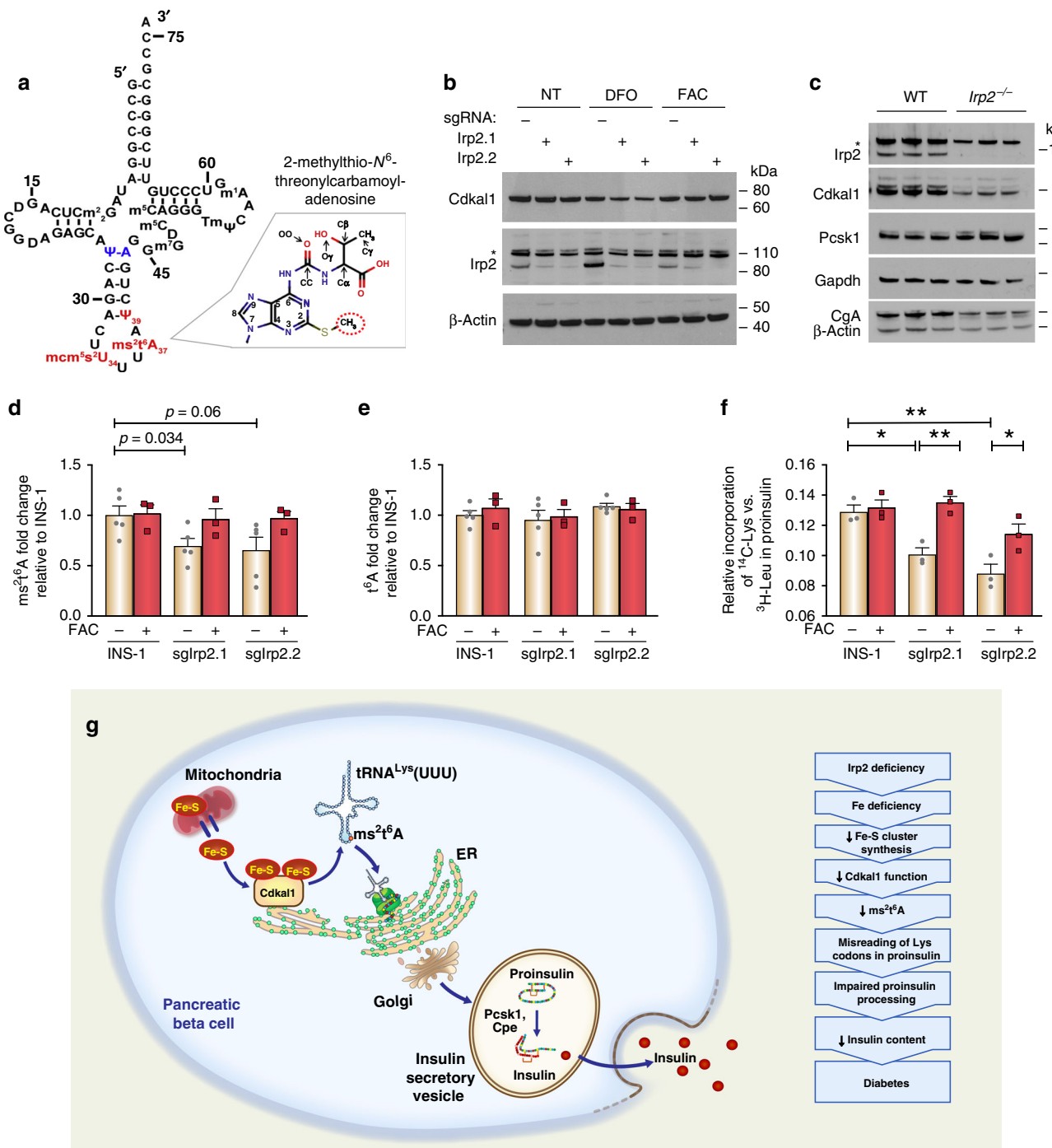

**Fig. 8 Irp2 deficiency reduces the ms2t6A modification in tRNA^Lys_UUU causing misreading of lysine codons in proinsulin. a** Secondary structure and sequence of tRNA^Lys_UUU with anticodon loop modifications 2-methylthio-N6-threonylcarbamoyl adenosine (ms2t6A37), methoxycarbonylmethyl-2-thiouridine (mcm5s2U34) and pseudouridine-39 (ψ−39). The ms2 group on ms2t6A37 is indicated by a dotted circle. Adapted from Vendeix, et al.[68]. **b** Western blot analysis of Cdkal1 in control, sgIrp2.1, and sgIrp2.2 INS-1 cells grown in medium with or without DFO or FAC for 18 h. β-Actin is a loading control. Asterisk, nonspecific band. **c** Western blot analysis of Irp2, Cdkal1, Pcsk1, and CgA in islets isolated from 6-month old WT and *Irp2^−/−* mice. Gapdh is a loading control for Irp2, Cdkal1 and Pcsk1 and β-actin is a loading control for CgA (*n* = 3 mice per genotype). **d**−**e** LC-MS analysis of ms2t6A (**d**) and t6A (**e**) modifications in control, sgIrp2.1, and sgIrp2.2 INS-1 cells grown in medium with or without supplemental FAC for 18 h (*n* ≥ 3 independent biological experiments). **f** Relative incorporation of ^14C-lysine versus ^3H-leucine in proinsulin immunoprecipitated from control, sgIrp2.1, and sgIrp2.2 INS-1 cells grown in medium with or without supplemental FAC (*n* = 3 independent biological experiments). Data in **d**, **e** are expressed as means ± s.e.m., unpaired two-tailed Student's *t* test, *$p < 0.05$, **$p < 0.01$ relative to control INS-1 cells; data in **f** are expressed as means ± s.e.m., compared by one-way ANOVA with Tukey's multiple comparison test, *$p < 0.05$, **$p < 0.01$. **g** Model of Irp2 regulation of proinsulin processing and insulin secretion in β cells. Source data are provided as a Source Data file.

sensitivity and secretion were not reported for these patients. We anticipate that our study will thus have relevance not only to β-cell function but also to other cell types and tissues that are functionally iron deficient. Our findings will also provide a foundation for further investigation of the role of iron deficiency in the pathogenesis of diabetes.

## Methods

**Animals.** Mice with global deficiency $Irp2^{-/-}$ mice were generated by inserting a self-excision cassette containing neomycin ($Neo^r$) linked to cre-recombinase ($Cre$) into exon 3 of the mouse $Irp2^{-/-}$ gene, as previously described[17]. This cassette (pACN) contains the Cre gene (driven by the testes-specific angiotensin-converting enzyme (tACE promoter) linked to $Neo^r$ (driven by the polymerase II promoter) and is flanked by loxP sites allowing for excision of $Neo^r$, as it passes through the male germ line. $Irp2^{-/-}$ mice were generated on a C57BL/6J and 129/Sv background, and backcrossed with C57BL/6J for five generations. $Irp2^{-/-}$ and WT littermates were obtained from intercrosses from $Irp2^{+/-}$ parents. Male and female mice from 2- to 18-months of age were used. Mice were kept in accordance with the recommendations in the Guide for the Care and Use of Laboratory Animals of the National Institutes of Health. The protocol was approved by the Institutional Animal Care and Use Committee (IACUC) of the University of Utah (Protocol Number: 18-12018). Mice were euthanized according to AVMA Guidelines for the Euthanasia of Animals. At the GMC, mice were maintained in IVC cages with free access to water and standard mouse chow containing 183 mg Fe/kg (Altromin no.1324, Altromin, Lage, Germany). All experiments and housing of the animals in the GMC were performed according to the German Law on the Protection of Animals by the Government of Upper Bavaria (Regierung von Oberbayern).

**Animal iron treatment.** Iron overload in mice was achieved by daily intraperitoneal injections of 1 mg of iron dextran (Sigma-Aldrich, cat # D8517) or PBS as a control for 5 consecutive days[59]. Mice were sacrificed, and pancreas and liver were harvested for iron content by ICP-MS, and pancreas for insulin and proinsulin measurements using ELISAs.

**Preparations for glucose clamp experiments.** To enable i.v. substrate infusion, a permanent silicone catheter was inserted into the left jugular vein under ketamine/xylazine anesthesia. For matching body fat and fat-free mass, mice were subjected to $^1$H-NMR analysis (MiniSpec, Bruker Optics Inc, Ettlingen, Germany) 6 days later. On the morning of the seventh postsurgical day, conscious mice were placed in restrainers (Opti-Lab, Munich, Germany) set on top of heating pads. The subcutaneously located catheter end, accessible via attached silk protruding from a small interscapular skin incision, was connected to 1cc syringes fastened in a microdialysis pump (CMA 402, Solna, Sweden). For immediate measurement of plasma glucose concentrations (Glucometer Ascensia Elite, Bayer, Leverkusen, Germany), tail-tip blood samples were collected in heparinized CB300 LH Microvettes. Plasma insulin concentrations were measured via an Ultrasensitive Mouse Insulin ELISA (Mercodia, cat#10-1249-01).

**Hyperglycemic clamp.** Experiments were carried out after 16 h fasting, 30.5 ± 0.3 weeks old, male WT ($n = 8$) and $Irp2^{-/-}$ ($n = 10$) mice. By means of a primed 10% intravenous glucose infusion, plasma glucose concentrations were acutely raised by adjusting the glucose infusion rate to reach and maintain target hyperglycemia (steady state, mmol/l: 17.9 ± 0.6 in WT vs. 18.5 ± 0.6 in $Irp2^{-/-}$). Steady state was defined as the last 45 min of the experiment. Blood samples were withdrawn from tail tips under basal conditions (min −10) as well as during the glucose clamp at min 5, 7.5, 10, 15, 20, 30, 45, 60, 75, 90, and 105 for the measurement of plasma glucose and insulin. At the end of the experiment, mice were anesthetized with an i.v. ketamine/xylazine overdose, and various organs were dissected and freeze-clamped. Statistical analyses were made using a two-tailed Student's $t$ test and for all hypotheses the significance level was $p < 0.05$. All data are expressed as mean ± s.e.m.

**Euglycemic-hyperinsulinemic clamp.** Experiments were carried out after 15 h fasting $Irp2^{-/-}$ ($n = 6$) and WT ($n = 5$) male mice at an age of 29.1 ± 0.1 weeks. By means of a primed-continuous insulin infusion (100 mU/kg min$^{-1}$ within 3 min, then 3.5 mU/kg min$^{-1}$, Humulin-R, Eli Lilly, Indianapolis), insulin levels were acutely raised and maintained at a physiological level until 120 min ("steady state" conditions, pmol/l: 372 ± 86 $Irp2^{-/-}$ vs. 374 ± 64 in WT), as previously described[60,61]. The resulting decline in plasma glucose concentration was counteracted and euglycemia maintained by a 20% glucose infusion applied at a variable rate. "Steady state" conditions were usually achieved within 60 min, and "steady state" defined as the last 30 min of the experiment ("steady state" plasma glucose, mmol/l: 8.1 ± 0.6 $Irp2^{-/-}$ vs. 8.2 ± 0.7 in WT). "Steady-state" glucose infusion rate (GINF) equals whole-body glucose utilization, and is considered an index of an organism's sensitivity to exogenous insulin. To assess whole-body glucose turnover, a continuous [3-$^3$H]glucose infusion (0.1 µCi/min, Biotrend, Cologne, Germany) was administered during a 120 min basal tracer equilibration period and continued throughout the subsequent euglycemic-hyperinsulinemic clamp. In order to allow estimation of insulin-stimulated glucose uptake in individual tissues, an i.v. injection of a 2-deoxy-D-[1-$^{14}$C]glucose bolus (10 µCi, Biotrend, Cologne, Germany) was administered at "steady-state" min 75. The experiment was terminated by an i.v. ketamine/xylazine injection, and tissues (liver, M. gastrocnemius and quadriceps, epididymal adipose tissue, heart) were immediately dissected and freeze-clamped. Plasma [3-$^3$H]glucose, $^3$H$_2$O (measurement of whole-body glycolysis rate) and 2-[$^{14}$C]DG concentrations between clamp min 77.5 and 120 and in the final 10 min of the basal period were measured in Somogyi filtrates. The rate of whole-body glucose turnover was calculated as the ratio of [3-$^3$H]glucose infusion rate (dpm/min) and plasma [3-$^3$H]glucose-specific activity (dpm/min µmol) during "steady-state". Hepatic [3-$^3$H]glucose production (HGP) was determined by subtracting the "steady-state" glucose infusion rate (GINF) from the rate of glucose whole-body turnover. Statistical analyses were made using a two-tailed Student's $t$ test, and for all hypotheses the significance level was $p < 0.05$. All data are expressed as means ± s.e.m.

**Hematology and clinical chemistry.** Basic hematological parameters were analyzed with a blood analyzer, which was validated for the analysis of mouse blood (ABC-Blutbild-Analyzer, Scil Animal Care Company GmbH; Viernheim, Germany), using the C57BL/6-mouse-chip-card. The number and size of red blood cells, white blood cells, and platelets were measured by electrical impedance and hemoglobin by spectrophotometry. Mean corpuscular volume (MCV), mean platelet volume (MPV), and red blood cell distribution width (RDW) were calculated directly from the cell volume measurements. The hematocrit (HCT) was assessed by multiplying the MCV with the red blood cell count. Mean corpuscular hemoglobin (MCH) and mean corpuscular hemoglobin concentrations (MCHC) were calculated from hemoglobin/red blood cell count (MCH) and hemoglobin/hematocrit (MCHC), respectively. Clinical chemistry parameters were measured using an Olympus AU 400 autoanalyzer, and adapted reagents from Olympus (Hamburg, Germany). Creatinine was measured using a kit from Biomed (Oberschleißheim, Germany) and NEFA (nonesterified fatty acids) using a kit of WAKO Chemicals (Neuss, Germany). Twenty-three different parameters were measured including various enzyme activities, and plasma concentrations of specific substrates and electrolytes. Besides the routinely applied clinical chemistry screen, unsaturated iron-binding capacity (UIBC) was determined and transferrin saturation calculated in these animals. Data were statistically analyzed with the level of significance set at $p < 0.05$ by ANOVA.

**Glucose and insulin tolerance tests.** For glucose tolerance tests, mice were fasted for 16–17 h, and baseline blood glucose was determined from tail-vein blood (Ascencia Elite XL Glucometer, Bayer Corp and the Accu Check Aviva, Roche). Conscious animals were then challenged with an intraperitoneal (i.p.) injection of glucose (2 g/kg body weight, Sigma-Aldrich), and blood glucose levels were determined at 15, 30, 60, 90, and 120 min post injection. Animals were tested at 10–12 weeks, 20–22 weeks, and 74–78 weeks of age. For insulin tolerance tests, random-fed conscious animals (20–22 weeks and 50–64 weeks of age) were injected with human recombinant insulin (0.75 U/kg body weight, Novolin R, Novo Nordisk). Tail-vein blood samples were taken before and 15, 30, 60, 90, 120, 150 and 180 min post injection.

**Insulin and proinsulin measurements.** For plasma insulin and proinsulin analyses, 100 µl of tail-vein blood was collected from overnight fasted mice for baseline insulin measurements, and 30 min post-i.p. glucose injection (2 g/kg body weight). For pancreatic insulin and proinsulin content, freshly harvested pancreata were incubated overnight in acid–ethanol mixture (1.5% HCl in 70% EtOH) at −20 °C. Tissue was homogenized and incubated overnight at −20 °C. Samples were centrifuged at 3000 × $g$ for 15 min at 4 °C. Supernatants were neutralized with equal volume of 1 M Tris-HCl pH 7.5 and centrifuged at 15,000 × $g$ for 10 min at 4 °C. Insulin and proinsulin were quantified in mouse pancreata, islets and plasma, and in INS-1 832/13 cells using a Mouse Insulin ELISA (ALPCO, cat# 80-INSMS-E01) and a Rat/Mouse Proinsulin ELISA (Mercodia, cat# 10-1232-01), respectively, and normalized to total lysate protein determined by Coomassie Plus Protein Assay Reagent (ThermoFisher, cat# 23200).

**Islet isolation and insulin and proinsulin secretion assays.** Mice were anesthetized using Avertin (0.015 mL/g body weight) and killed by cervical dislocation. Pancreata were perfused through the common bile duct with 9 mL of Hanks balanced salt solution (HBSS, Invitrogen) containing 0.1 mg/mL DNase I (Roche), 25 mM HEPES, and 0.3 mg/mL Liberase RI Purified Enzyme Blend (Roche). Pancreata were extracted and incubated in a 37 °C water bath for ~12 min in HBSS/Liberase RI solution. Pancreatic tissue was disrupted by vigorous shaking for 10 s and washed twice with HBSS containing 0.1 mg/mL DNase I and 10% heat inactive fetal bovine serum (FBS, Invitrogen). Islets were handpicked twice from exocrine tissue. For glucose-stimulated insulin and proinsulin assays, ten islets were size-matched and handpicked to a 96-well plate and incubated in Krebs–Ringer bicarbonate (KRB)-HEPES buffer (10 mM HEPES, pH 7.35, 140 mM NaCl, 3.6 mM KCl, 0.5 mM NaH$_2$PO$_4$, 0.5 mM MgSO$_4$, 1.5 mM CaCl$_2$, 2 mM NaHCO$_3$, 0.1% bovine serum albumin (BSA), 2.5 mM glucose) for 1 h at 37 °C. Triplicate

assays were performed for each mouse. Islets were then treated with 16.7 mM glucose for an additional 1 h at 37 °C. Islets were sonicated in 1 mL HBSS at settling 4 with $10 \times 1$ s bursts (Brinkmann Sonic Dismembrator, Fisher Scientific) and lysates used for insulin and proinsulin quantitation.

For iron and DFO experiments, freshly isolated islets were assessed for insulin and proinsulin secretion in KRB–HEPES buffer containing 2.5 mM glucose, as described above. Islets were then cultured overnight in RPMI-1640 medium supplemented with 10% fetal calf serum, 10 mM HEPES, 2 mM L-glutamine, 1 mM sodium pyruvate, 0.05% β-mercaptoethanol (β-ME), and 5% gentamicin along with DFO (50 μM) or FAC (50 μg/mL) at 37 °C with 5% $CO_2$. Islets were washed and then incubated in KRB–HEPES buffer supplemented with 16.7 mM glucose for an additional 1 h at 37 °C. Supernatants were collected and islets sonicated as described above.

For glucose-stimulated insulin and proinsulin secretion assays in EV and shIrp2 expressing INS-1 832/13 cells, ~0.05 × $10^6$ cells were plated onto a 24-well plate and grown to 100% confluency. The standard culture medium containing 11 mM glucose was changed to medium containing 5 mM glucose 18 h before performing secretion assays[62]. Cells were washed and preincubated for 2 h in HBSS (20 mM HEPES, pH 7.2 with 114 mM NaCl, 4.7 mM KCl, 1.2 mM $KH_2PO_4$, 1.16 mM $MgSO_4$, 2.5 mM $CaCl_2$, 25.5 mM $NaHCO_3$ and 0.2% BSA). Insulin and proinsulin secretion were measured by static incubation in 0.8 mL of HBSS containing 5 mM or 15 mM glucose for a 2 h-period. For all experiments, insulin and proinsulin secretion and content were assayed by the Mouse Insulin ELISA and the Rat-Mouse Proinsulin ELISA, respectively, and normalized to total lysate protein using Coomassie Plus Protein Assay Reagent.

**Generation of stable Irp2-deficient insulinoma cell lines.** The rat insulinoma INS-1 832/13 cell line was kindly provided by Dr. Chris Newgard (Duke University). Cells were cultured in the RPMI-1640 medium supplemented with 10% fetal calf serum, 10 mM HEPES, 2 mM L-glutamine, 1 mM sodium pyruvate, 0.05 β-ME and 5.0 μg/mL puromycin, and grown at 37 °C with 5% $CO_2$, as described[62]. For Irp2 depletion by short-hairpin RNA (shIrp2 RNA), INS-1 832/13 cells were stably transduced with empty vector (EV) (pLentiLox3.7puro) or shIrp2 RNA (pLentiLox3.7puro::IRP2-4)[63] kindly provided by Dr. Othon Iliopoulos (Massachusetts General Hospital Cancer Center). The sequences for shIrp2 RNAs are:

Irp2-forward: 5′-TGGATTCTGGGGTGGGGGGGTCTCTTCACCCCCCAC CCCAGAATCCTTTTTTC-3′

Irp2-reverse: 5′-TCGAGAAAAAAGGATTCTGGGGTGGGGGGTGAAGAG ACCCCCCCACCCCAGAATCCA-3′

For CRISPR/Cas9-targeted depletion of Irp2, two distinct short guide RNAs (sgRNAs) against rat Irp2 E3-S7 (sgIrp2.1) and E5-S14 (sgIrp2.2) were obtained from the Mutation Generation and Detection Core (University of Utah) and used to generate sense and antisense oligonucleotides containing 5′-CACC and 3′-CAAA overhangs, respectively. The sgRNAs were annealed and ligated into CRISPR-Cas9 vector pSpCas9(BB)-2A- Puro (PX459) V2.0 provided by Feng Zhang (Addgene plasmid # 62988)[64]. Irp2 sgRNA sequences are:

sgIrp2.1-forward: 5′-GCTTGAAGAACAACACGGGCG-3′
sgIrp2.1-reverse: 5′-CGAACTTCTTGTTGTGCCCGC-3′
sgIrp2.2-forward: 5′-GCGAGGCCAGACTACCTGCCG-3′
sgIrp2.2-reverse: 5′-CGCTCCGGTCTGATGGACGGC-3′.

INS-1 832/13 cells were grown to 70–90% confluency in complete RPMI-1640 medium on 60-mm tissue culture plates. Cells were thoroughly washed with 1× Dulbecco's phosphate-buffered saline (DPBS, ThermoFisher cat# 14190250) and transfected with 20 μg of sgIrp2.1 or sgIrp2.2 plasmid constructs containing puromycin resistance using Lipofectamine 2000 (ThermoFisher, cat# 11668027). Transfected cells were selected 48 h later with 5 μg/mL puromycin dihydrochloride. Puromycin resistant cells were assayed for loss of Irp2 by western blot analysis.

**Western blot analysis.** Isolated islets (100–150) were lysed in Triton Lysis Buffer (10 mM Tris-HCl pH 7.4, 150 mM NaCl, 10 mM EDTA, 1% Triton X-100, 1 mM DTT, and Halt Protease & Phosphatase Inhibitor Cocktail (ThermoFisher, cat #78446) using a micro-dounce. Lysates were cleared by centrifugation, and protein concentration was determined using Coomassie Plus Protein Assay Reagent. Islet lysates were boiled in lithium dodecyl sulfate (LDS) sample buffer (Invitrogen) containing 2.5% β-ME and analyzed by NuPAGE™4–12% Bis-Tris gels (Invitrogen) with MES SDS-running buffer. INS-1 832/13 cells expressing EV, shIrp2, sgIrp2.1, or sgIrp2.2 were treated with ferric ammonium citrate FAC (10–50 μg/mL) or the iron chelator desferrioxamine (50 μM) for 18 h, and lysed in Triton Lysis Buffer and cell lysates (15–30 μg protein) were analysis by 4–12% Bis-Tris gels. Proteins were transferred to a Hybond-ECL nitrocellulose membrane (Amersham) and probed with the following antibodies: chicken anti-Irp1 polyclonal antibody (PAb) (1:5000)[65]; rabbit anti-Irp2 PAb antibody (1:2000)[65]; rabbit anti-ferritin PAb (UT106, custom made Covance, 1:3000); mouse TfR1 monoclonal antibody H68.4 MAb (ThermoFisher, cat# 16-6800, 1:1000); rabbit Cdkal1 PAb (ThermoFisher, cat# PA5-29077, 1:1000); rabbit PC1/3 (Pcsk1) PAb (Millipore, cat# AB10553, 1:1000); mouse chromogranin A SP12 MAb (ThermoFisher cat# MA5-14536); mouse eIF-2α MAb (ThermoFisher, cat# AHO0802, 1:2000); rabbit phospho-eIF2α Ser52 PAb (ThermoFisher, cat# 44-728 G, 1:500); rabbit BiP/Grp78 PAb (ThermoFisher, cat# PA1-014A, 1:2000); mouse β-actin MAb

(ThermoFisher, cat# 8H10D10, 1:2000); and mouse GAPDH MAb (ThermoFisher, cat# MA5-15738, 1:2000). Horseradish peroxidase-conjugated secondary antibodies were bound, and proteins were visualized using Western Lighting Chemiluminescence Reagent Plus (PerkinElmer Life Sciences). Membranes were stripped for 10 min at 65 °C in stripping buffer (62.5 mM Tris-HCl, pH 6.8, 100 mM β-ME and 2% SDS) or for 5–15 min at 37 °C in Restore™ PLUS Western Blot Stripping Buffer (ThermoFisher, cat# 46430). Uncropped and unprocessed scans are located in the Source Data file.

**RNA-electrophoretic mobility shift assays.** RNA-EMSAs were performed by incubating whole-cell lysates (15 μg) with a [$^{32}$P]GTP-labeled ferritin L-IRE probe in 10 mM HEPES, pH 7.6, 3 mM $MgCl_2$, 40 mM KCl, 5% glycerol, 1 mM DTT for 20 min at room temperature[66]. Heparin (Sigma) (50 μg/μl) and RNase T1 (Roche) (1 U/μl) were added simultaneously to the lysates for 10 min. Irp1- and Irp2-RNA complexes were resolved by 5% non-denaturing PAGE (acrylamide:methylene bisacrylamide ratio, 60:1), and the dried gel exposed to a PhosphorImager screen for analysis.

**Tissue metal content measurement.** Metal content was determined by digesting 20–30 mg of pancreas or liver or 100–200 isolated islets in 40% metal-free nitric acid at 95 °C. Samples were diluted in water and analyzed by PerkinElmer Optima 3100XL inductively coupled plasma optical emission spectrometer (ICP- OES) for islets and Agilent 7900-ICP (inductively coupled plasma mass spectrometer (ICP-MS)) for liver and pancreas. Metal concentrations were corrected for dilution and normalized to the total protein or number of islets. Iron content in shIrp2, sgIrp2.1, sgIrp2.2, and control INS-1 832/13 cell lines was determined by ICP-MS. Iron content was normalized to total protein.

**Immunohistochemistry and immunofluorescence.** Pancreata were fixed with 4% buffered formalin immediately after dissection and embedded in paraffin. Pancreas sections (5 μm) were rehydrated with xylene followed by decreasing concentrations of ethanol, and antigen retrieval was performed using Target Retrieval Solution (Dako, cat# S1699) at 90 °C for 30 min followed by cooling at room temperature for 20 min. Sections were incubated in blocking solution (5% donkey serum, 1% BSA) for 1 h at room temperature and then stained with guinea pig anti-insulin PAb (Dako, cat# A0564, 1:200) for 16 h. Sections were washed in PBS and incubated with Peroxidase AffiniPure Donkey anti-guinea pig IgG secondary antibody (Jackson ImmunoResearch, cat# C706-035-48, 1:500) for 1 h and visualized using by 3'3'-diaminobenzidine (DAB). For islet morphometric analysis, sections were taken at 120 um intervals. Images (4×) spanning the entire pancreas were acquired using a Nikon Widefield CCD/Spinning Disk microscope that automatically stitched images together to generate a compound image. Image J software was used to quantify islet area and the fraction of insulin-stained tissue area per pancreas for 2.5-, 7.5-, and 18-month-old mice ($n = 3$–6 mice per genotype). Five cell clusters were considered to be an islet. β-cell mass was calculated by multiplying the fraction of insulin-stained tissue area by pancreas weight. Hematoxylin and eosin (H&E) staining of paraffin-embedded pancreatic sections was performed by the Huntsman Cancer Biorepository and Molecular Pathology Shared Resource (University of Utah).

For double-immunofluorescence studies, pancreas sections were deparaffinzed and antigen retrieval was performed as described above. Sections were incubated in TBST blocking solution (Tris-buffered saline containing 0.1% Tween) containing 1% BSA for 1 h at room temperature and then incubated with the following antibodies in TBST at 4 °C for 16 h: rabbit anti-ferritin PAb (UT106, 1:100), mouse anti-TfR1 MAb (H68.4, ThermoFisher, cat# 13-6800, 1:250), rabbit anti-Glut2 PAb (Millipore, cat# 07-1402, 1:200). Sections were washed in TBST, costained with guinea pig anti-insulin PAb (Dako, cat# A0564, 1:200) for 1 h, washed in PBS and incubated with secondary antibodies Alexa Fluor™ 546 goat anti-guinea pig (ThermoFisher, A-11074, 10 μg/ml) along with Alexa Fluor™ 488 goat anti-rabbit (ThermoFisher, cat# A-11034, 10 μg/ml) or Alexa Fluor™ 488 goat anti-mouse (ThermoFisher, A-11001, 10 μg/ml) in TBST plus 1% BSA for 1 h at room temperature. Images were acquired using an Olympus FV1000 confocal microscope at the same time using identical camera settings.

**Quantitative real-time RT-PCR (qPCR).** The total RNA was extracted from islets isolated from 39–49-week-old WT and $Irp2^{-/-}$ mice using TRIzol reagent (Invitrogen). cDNA synthesis was performed with total RNA (0.5–1 μg) using Super-Script III First-Strand synthesis SuperMix for qPCR (Invitrogen). qPCR was performed on an ABI Prism 7900HT Sequence Detection System using TaqMan master mix and TaqMan Gene Expression Assays: $Ins1$ Mm01259683_g1; $Ins2$ Mm00731595_gH; $Fthl1$ Mm00850707_g1; $Ftl1$ Mm03030144_g1 and $Tfrc$ Mm00441947_g1 (ThermoFisher). The ΔΔCT method was used to determine mRNA-fold change. All experiments were performed using 5–6 mice per genotype with duplicate technical replicates, and fold changes were normalized to $actb$ levels.

**Enzyme activity and ATP measurements.** Complex I and IV activities were measured in INS-1 832/13 cell extracts using Complex I Enzyme Activity Dipstick Assay Kit (Abcam, cat# ab109720) and Complex IV Rodent Enzyme Microplate Assay Kit (Abcam, cat# ab109911) following the manufacturer's protocols. Whole-

cell extracts (25 µg for Complex I assay and 50 µg for Complex IV assay) were used. Aconitase activity was measured in INS-1 832/13 cells enriched mitochondrial and cytosolic extracts (Mitochondria Isolation Kit for Cultured Cells, ThermoFisher, cat# 89874) using the Aconitase Enzyme Activity Microplate Assay Kit assay (Abcam, cat# 109712). Cell lysates ($n = 4$ independent biological experiments) were stored at $-80\,°C$ and then assayed together. For measurement of intracellular ATP, INS-1 832/13 cells were preincubated in HBSS (secretion buffer) with 2.5 mM glucose for 30 min at 37 °C. Cells were then incubated with 5 mM glucose or with 16.7 mM glucose for 1 h. ATP content was measured in triplicate samples using the ATP Determination Kit (Molecular Probes, cat# A22066) and normalized to total cellular protein. For enzyme activity and ATP assays, cells were treated with FAC (10 µg/mL) for 18 h before performing the assay.

**Mass spectrometric analysis of the ms²t⁶A modification**. All chemicals and reagents used were obtained at the highest purity available and were used without further purification unless stated. Benzonase, calf intestinal alkaline phosphatase, butylated hydroxytoluene, acetonitrile, and buffer salts were purchased from Sigma-Aldrich. Coformycin was obtained from the National Cancer Institute. Phosphodiesterase I was purchased from (Worthington cat# LS003926). Water purified through a Milli-Q system (Millipore) was used throughout our studies. VWR-brand 10 kDa MWCO centrifugal filters were used for digestion clean up. *Small RNA isolation*: RNA was isolated from parental, sgIrp2.1, and sgIrp2.2 INS-1 832/13 cells ($4 \times 10^6$ cells) grown for 18 h in medium with or without supplemented FAC (10 µg/mL). RNA isolation was performed using miRNA Pure Link kit according to the manufacturer's instructions (ThermoFisher, cat# K157001). RNA was resuspended in RNASE/DNASE-free water, and TapeStation was used to determine RNA quality and concentration. *Nucleoside preparation*: Small RNA (1–2 µg) from each sample was digested using a mixture of Benzonase (0.075 U), phosphodiesterase I (0.001 U), and calf intestinal alkaline phosphatase (0.17 U) in a final reaction volume of 50 µl. The reaction was supplemented with MgCl₂ at a final concentration of 5 mM. Coformycin (NCI), a nucleobase deaminase inhibitor, was added at a concentration of 1 µg/mL, and butylated hydroxytoluene (an anti-oxidant) was included at a concentration of 0.3 mM. The digestion was allowed to proceed for 2 h at 37 °C, and was stopped upon removal of the enzymes by microfiltration with 10 kDa spin filters. *LC-MS analysis of RNA nucleosides*: Ribonucleosides were resolved using a Synergy Fusion RP (2.5 µm, 100 Å, 100 × 2 mm) HPLC column (Phenomenex), on an Agilent 1290 HPLC system equipped with a diode array detector. Mobile phase A was 5 mM ammonium acetate, and mobile phase B was pure acetonitrile. Gradient elution started with 100% A for 1 min, increased to 10% B after 10 min, 40% after 14 min, 80% after 20 min and regeneration of starting conditions with 100% A for 5 additional min. The flow rate was 0.35 mL/min, and the column temperature 35 °C. The effluent from the column was directed through the DAD before entering the Agilent 6490 triple quadrupole mass spectrometer in dynamic multiple reaction monitoring (MRM) mode. The MS was operated in positive ion mode with the following parameters: electro-spray ionization (ESI-MS), fragmentor voltage 380 V, cell accelerator voltage 7 V, N₂-gas temperature 250 °C, N₂ gas flow 11 L/min, nebulizer 20 psi, capillary 1800 V. Modified nucleosides were identified based on having both the correct retention time and mass transition. Under our chromatography conditions, ms²t⁶A eluted at 12.5 min with a mass transition of $m/z$ 459 → 327, corresponding to the neutral loss of ribose. *Data analysis*: In each sample, the MS peak areas for the four canonical ribonucleosides (cytidine, uridine, guanosine, and adenosine) were integrated and summed. Peaks corresponding to the transitions for modified nucleosides were normalized to the summed peak areas of the canonical nucleosides to account for the amount of injected RNA. The adjusted peak areas of the knockout and iron-supplemented samples were then calculated as a percentage of the INS-1 832/13 control sample.

**Metabolic ¹⁴C-lysine and ³H-leucine labeling**. Parental control, sgIrp2.1, and sgIrp2.2 INS-1 832/13 cells were grown to 90% confluence in medium with or without supplemented FAC (10 µg/mL) for 18 h. Cells were then incubated with KRB containing 0.1% BSA and 2.8 mM glucose for 30 min at 37 °C. The buffer was replaced with KRB containing 0.1% BSA and 16.7 mM glucose along with 10 µl of L-[¹⁴C(U)-lysine (1 µCi/reaction) (PerkinElmer, cat# NEC280E050UC) and 10 µl of L-[3,4,5-³H(N)-leucine (10 µCi/reaction) (PerkinElmer, cat# NET460250UC) and incubated at 37 °C for 1 h. Cells were washed twice with PBS and then lysed in T-PER™ Tissue Protein Extraction Reagent (ThermoFisher, cat# 78510). Protein content was measured by the Coomassie Plus Protein Assay Reagent, and 500 µg of protein from each sample was incubated with Dynabeads™ Protein A (Thermo-Fisher, cat# 10001D) previously loaded with 5 µl proinsulin/insulin LB610 MAb (Cell Signaling, cat# 8138). The beads were washed, and proinsulin was released from the beads with 20 µl of elution buffer (50 mM glycine pH 2.8) plus 10 µl of LDS sample buffer by heating the samples at 70 °C for 10 min. The samples were diluted 1:1000 (5 µl/5000 µl) in scintillation fluid, and radioactivity measured in triplicate. The results are expressed as relative incorporation of ¹⁴C-lysine versus ³H-leucine in proinsulin.

**Statistical analysis**. All experiments were performed using at least three independent biological replicates that reflect different sources of material. Data are

expressed as means ± standard error (s.e.m.). For two group comparison, an unpaired two-tailed Student's $t$ test was used. For multiple group comparisons, one-way ANOVA was used followed by Tukey's multiple comparison test. For all hypotheses, the significance level was $p < 0.05$. Statistical analyses were performed using GraphPad Prism 7.0, 7.04 and 8.30, and Excel 2013.

## Data availability
The source data underlying Figs. 1a–g, 2a–e, 3a–n, 4a,b,e, 5a–i, 6a, c–i, 7a–h, 8b–f, and Table 1 and unprocessed gel scans are provided as a Source Data file. All other data supporting the findings of this study are available from the corresponding author upon reasonable request.

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

## Acknowledgements

This work was supported by National Institute of Health (NIH) grants R01GM045201 and R01DK107712, and University of Utah SEED grant (to E.A.L.); NIH grants R01GM070641, R01ES026856, R01ES024615 (to P.C.D.); the National Research Foundation of Singapore through the Singapore-MIT Alliance for Research and Technology (to. P.C.D.); and SMA3 Programme Fellowship (to W.M.C.); Hematology Training Program grant T32DK007115 (to C.P.A. and K.B.Z.); NIEHS Toxicology Training grant T32ES007020 (to J.H.); and American Diabetes Association Minority Postdoctoral Fellowship Award 1-18-PMF-020 (to M.C.F.S.). This work was supported by NIDDK Cooperative Hematology Specialized Core Center grant 1U54DK110858, specifically the Mutation Generation and Detection Core and Iron and Heme Core; by the German Federal Ministry of Education and Research Infrafrontier grant 01KX1012 (to M.H.A) and the German Center for Diabetes Research (D.Z.D.). The analysis of t6A and ms2t6A was performed in part in the Bioanalytical Core Facility of the MIT Center for Environmental Health Sciences, which is supported by NIH grant P30ES002109. We thank Ms. Diana Lim for figure preparation.

## Author contributions

Conceived the experiments and wrote the paper: E.A.L., M.C.F.S., C.P.A., S.N., K.B.Z., and P.C.D. Supervised the study at the GMC: V.G., H.F., E.W., J.R. and M.H.A. Performed experiments: E.A.L., M.C.F.S., C.P.A., S.N., K.B.Z., S.J.R., M.K.S., B.R., J.R., W.M.C., M.R. and J.H. Analyzed the data: E.A.L., M.C.F.S., C.P.A., S.N., K.B.Z., S.J.R., B.R., J.R., W.M.C., J.H. and P.C.D. Writing, reviewing, and editing: E.A.L., M.C.F.S., S.N., K.B.Z., J.H. and P.C.D.

## Competing interests

The authors declare no competing interests.
