## [Peer Review File · Nature Communications]

Reviewers' comments:

Reviewer #1 (Remarks to the Author):

This is a complete and compelling paper providing evidence for Cdkal1-mediated glucose intolerance in the setting of functional iron deficiency due to mutation of Ireb2. It provides an informative extension of earlier findings [Wei FY et al. (2011) JCI 121:3598-3608] and introduces a role for iron in modification of Cdkal1 activity. I have few suggestions, none major.

1. There is a single case report of a patient with microcytic anemia and diabetes due to an anti-transferrin receptor antibody [see Hyman ES (1983) Lancet 1(8316):91-5 and Larrick JW, Hyman ES (1984) N Engl J Med 26:214-8]. It seems highly likely that the etiology of her diabetes is explained by the authors' findings and this could be mentioned in the Discussion.
2. When impaired Cdkal1 function causes defective modification of the tRNA and consequent misreading of lysine codons is another amino acid inserted in place of lysine, or is the protein frameshifted/truncated? If the latter, wouldn't the amount of pro-insulin be decreased, or at least the amount of normal proinsulin?
3. The description of Fig. 4a doesn't make sense to me. First, there doesn't seem to be any Fth1 in WT cells—is that correct? The amount of Fth1 appears to be increased in Irp2^{-/-} islets but the text in line 159 describes them as decreased. Similarly, for Fig. 4b the text in line 162 describes the level of Fth1 mRNA as increased in Irp2^{-/-} islets but the figure shows it to be decreased. There also appear to be errors in the sentence “Fth1 mRNA levels were similar . . . , but, unexpectedly, Fth1 protein levels increased, which may be a transcriptional response to compensate for increased Fth1 protein (Fig. 4b).”
4. Figure 5a is reported to show a decrease in Tfr1 in cells treated with shIrp2 but that is not apparent in the second lane of the western blot. Could the results be quantitated, or is there a better blot?

Nancy Andrews

Reviewer #2 (Remarks to the Author):

The present study investigated the role of Irp2 in the regulation of insulin biosynthesis and glucose metabolism. Irp2-knockout mice exhibited glucose intolerance due to an impairment in insulin secretion from pancreatic beta-cells. The decrease of insulin secretion was resulted from an increase in the proinsulin content and the decrease in the total insulin content in the beta-cells.

Mechanistically, Irp2-deficiency suppressed the mRNA and protein levels of transferrin receptor (TfR1) and upregulated the protein level of ferritin, leading to the decrease in the total iron content in the pancreatic beta-cells. The decrease of intracellular iron subsequently impaired the activity of Cdkal1, an iron-sulfur cluster-dependent enzyme required for the methylthio-modification of tRNA^{Lys}(UUU). Dysregulation of Cdkal1 then impaired proinsulin translation at Lys codon. Importantly, supplement of iron in Irp2-deficient cells enhanced Cdkal1 activity, which improved the proinsulin translation fidelity and increased insulin contents. These results demonstrate that Irp2 is essential for the iron homeostasis and controls translational fidelity of proinsulin via tRNA modification enzyme Cdkal1. Overall, these findings are interesting and novel.

Major concerns:

1. The 5-month old Irp2-knockout mice showed glucose intolerance after intraperitoneal (i.p.) injection of glucose (Figure 1a), but the plasma insulin levels of knockout mice did not differ from wild-type mice at the basal level and after the i.p. injection of glucose (Figure 2a). The plasma insulin data was not consistent with the hyperglycemic clamp data (Figure 2d), which showed that insulin secretion was markedly decreased in Irp2-knockout mice. Authors should explain this discrepancy.
2. The finding that Irp2-deficiency resulted in the accumulation of proinsulin is particularly important in this study. Authors should examine the plasma proinsulin level or plasma proinsulin/insulin ratio in Irp2-deficient mice with i.p. administration of glucose. Authors also need to examine the proinsulin secretion in isolated islets after glucose stimulation.

Minor concerns:

1. Figure 3e and 3f. Authors showed that the average islet area and the beta-cell mass are comparable between Irp2-knockout mice and wild-type mice. HE staining of pancreas and islets need to be presented to support these data.
2. Figure 3. There are two “b” panels. The upper right panel should be panel “c”.
3. Figure 4d. Authors claimed that TfR1 staining was reduced in Irp2-knockout beta-cells (page 7, line 164). This is not convincing because the quality of the TfR1 staining is very poor. A majority of beta-cells was not stained by the antibody even in the wild-type islets.
4. Figure 5a. Authors claimed that TfR1 was modestly reduced in Irp2-knockdown INS cells (page 8, line 175). This claim is not correct because the intensity of TfR1 band in Irp2 knockdown cells was comparable to the TfR1 band in control cells (lane 1 versus lane 2). Please do not overstate the result, or author should provide the quantitative data.

5. Figure 5b. Please provide the method for this experiment.
6. Figure 5c. In addition to Calcein-AM, authors need to use ICP-MS to measure the iron content in Irp2-knockdown INS cells (figure 5) and Irp-knockout INS cells (figure 6).
7. Page 7, line 145: It is not appropriate to state that there is no general defect in ER protein maturation just because Glut2 showed no change between Irp2-knockout islets and wild-type islets. Please rephrase.
8. Page 7, line 159: "... Fth1 levels decreased in Irp2^{-/-} islets...". Should this be "... Fth1 levels increased in Irp2^{-/-} islets..."?
9. Page 7, line 162: "... Fth1 mRNA levels increased...". Should this be "... Fth1 mRNA levels decreased..."?
10. Page 8, line 189: "...from reduced insulin content...". Should this be "...from increased insulin content..."?
11. Page 13, line 362. "min 120" should be "120 min".
12. The units for glucose and insulin are not consistent. For example, insulin is shown as ng/ml in figure 2b, but is shown as pmol/l in figure 2d. Please be consistent.
13. Please provide sequencing data to show how Irp2 gene was edited in the two lines of Irp2-knockout cells

Reviewer #3 (Remarks to the Author):

The main point of this paper is that beta cell deficiency of labile iron perturbs the fidelity of proinsulin translation leading to diabetes. Specifically, in this manuscript, the authors utilize mice with whole body knockout of the RNA-binding protein Irf2, and show that these mice develop glucose intolerance with an increase in intracellular proinsulin and a decrease in intracellular insulin. They indicate that Irf2 loss of function leads to increased sequestered iron but decreased labile iron that they link with destabilization of the Fe-S enzyme Cdkal1 (itself a T2D susceptibility gene), thus resulting in improper translation of lysine codons in proinsulin that could account for impaired proinsulin processing and possible beta cell ER stress. Using Irf2 deficient INS1 cells as a model, some of these adverse effects are improved by preincubation for 18 hours with ferric ammonium citrate (FAC) to increase labile iron. The connection of Irf2 levels to regulation of Cdkal1 to ultimately drive regulation of proinsulin processing through methylation and proper lysine codon translation is novel, and the story is well written. However, a major concern is insufficient validation and the lack of more physiological evidence to support the mechanism that they are hypothesizing. At this point, additional studies are necessary to support the conclusions and enhance the significance of the manuscript.

Specific Comments:

1. The authors show that 18 hours of FAC supplementation rescues insulin-related phenotypes in Irf2-deficient beta cells. To demonstrate physiological significance, the authors should try to determine if iron supplementation (even a relatively brief course to prevent toxicity) can restore insulin production or secretion in vivo.
2. Independent of #1, the authors should determine if FAC is capable to restore insulin processing and Cdkal1 levels in isolated Irf2 null islets instead of relying solely on results from cell lines.
3. Immunofluorescence microscopy with co-stained organelle markers, and transmission electron microscopy of the islets of Irf2 null pancreata are needed to understand where in beta cells the increased proinsulin may be accumulating.
4. Many of the mechanistic connections drawn between Irf2 and impaired insulin processing may be linked by effects on Cdkal1, but some attempts should be made to exclude other possibilities.
 - a. Fe-S clusters regulate key mitochondrial proteins and proinsulin intracellular transport and processing is an energy intensive process. Low Complex I activity in shIrf2 INS-1 cells could lead to the generation of ROS, which could then lead to untoward downstream effects. Thus, pharmacologic approaches to enhance mitochondrial fuel utilization (such as mitochondrial targeted substrates including methylpyruvate or amino acids leucine/glutamine/alanine) and/or reduce ROS (antioxidants) independent of using an iron supplement are needed to determine if it is truly Cdkal1 deficiency that impairs insulin processing in Irf2 null islets or simply bioenergetic failure.
 - b. Can overexpression of Cdkal1 more efficiently rescue proinsulin processing deficits in Irf2 null cells plus and minus FAC?
5. As best I can tell, the Irf2 null mice have been around for 15 years, but implicating deficiency of Irf2 directly in human disease has not really happened yet. As far as I know, Irf2 is not even on the long list of diabetes GWAS candidates. With this in mind, it would be best if the authors would attempt to demonstrate a proinsulin processing defect in primary human islets with deficiency of labile iron, perhaps by utilizing iron chelation on the isolated islets.
6. The activation of eIF2a may reflect ER stress; additionally iron status might independently affect eIF2a phosphorylation – these points are poorly developed. At minimum, the authors should re-check phospho-eIF2a levels in IRP2 deficient cells treated \pm FAC, as well as a similar experiment after Cdkal1 overexpression.

Minor:

7. The Figure 6G legend was omitted.
8. The FAC abbreviation on line 473 is misplaced in the sentence.

Response to reviewers' comments:

We appreciate the thorough critique of our manuscript that we found insightful and constructive. Based on the reviewers' comments and suggestions, we have revised the manuscript. New experiments and responses to reviewers' comments are described below.

New experiments:

Fig. 2a: Plasma insulin levels after an i.p. glucose injection in 7-month old WT and *Irp2*^{-/-} mice

Fig. 3d: Plasma proinsulin-to-insulin ratio in 7.5-month old WT and *Irp2*^{-/-} mice

Fig. 3l,n: Proinsulin secretion measured in WT and *Irp2*^{-/-} islets

Fig. 5a: Western blot to show reduced TfR1 in shIrp2 INS-1 cells

Fig. 5c: Total iron content in shIrp2 cells determined by ICP-MS

Fig. 5j-m: Iron-overload in *Irp2*^{-/-} mice to show normalization of insulin and proinsulin content

Fig. 6b: Total iron content in sgIrp2.1 and sgIrp2.2 cells determined by ICP-MS

Supplementary Fig. 3a-d: Hematoxylin & eosin and insulin staining of WT and *Irp2*^{-/-} pancreata to show normal morphology

Reviewer #1 (Remarks to the Author):

This is a complete and compelling paper providing evidence for Cdkal1-mediated glucose intolerance in the setting of functional iron deficiency due to mutation of *Ireb2*. It provides an informative extension of earlier findings [Wei FY et al. (2011) JCI 121:3598-3608] and introduces a role for iron in modification of Cdkal1 activity. I have few suggestions, none major.

1. There is a single case report of a patient with microcytic anemia and diabetes due to an anti-transferrin receptor antibody [see Hyman ES (1983) Lancet 1(8316):91-5 and Larrick JW, Hyman ES (1984) N Engl J Med 26:214-8]. It seems highly likely that the etiology of her diabetes is explained by the authors' findings and this could be mentioned in the Discussion.

Response: These references are included in the Discussion.

2. When impaired Cdkal1 function causes defective modification of the tRNA and consequent misreading of lysine codons is another amino acid inserted in place of lysine, or is the protein frameshifted/truncated? If the latter, wouldn't the amount of pro-insulin be decreased, or at least the amount of normal proinsulin?

Response: Wei et al. (JCI, 121:3598, 2011) reported that loss of Cdkal1 function causes the insertion of another amino acid rather than producing a frameshift/truncated protein. They used a dual luciferase assay to detect frameshifts occurring during decoding of AAA and AAG in *Bacillus subtilis* wildtype and *ΔygeV* (Cdkal1 bacterial ortholog) and found no significant frameshift activity in the *ΔygeV* strain. They found, however, that when constructs were induced with IPTG, translation of AAA and AAG was reduced. They concluded that the m^{s2}t^{6A} modification is critical to prevent misreading of lysine codons when protein synthesis is high. We discussed this study in the Discussion. We are currently working with a colleague at the university who has experience in mass spec analysis of RNA modifications to identify the amino acid inserted in PI in place of lysine.

3. The description of Fig. 4a doesn't make sense to me. First, there doesn't seem to be any Fth1 in WT cells – is that correct? The amount of Fth1 appears to be increased in *Irp2*^{-/-} islets but the text in line 159 describes them as decreased. Similarly, for Fig. 4b the text in line 162 describes the level of Fth1 mRNA as increased in *Irp2*^{-/-} islets but the figure shows it to be decreased. There also appear to be errors in the sentence "Ftl1 mRNA levels were similar ..., but, unexpectedly, Fth1 mRNA levels increased, which may be a transcriptional response to compensate for increased Fth1 protein (Fig. 4b)."

Response: Sentences were corrected.

"Ftl1 was similarly expressed in WT and *Irp2*^{-/-} islets, while Fth1 was not detected in WT islets, but was notably increased in *Irp2*^{-/-} islets (Fig. 4a)."

"Ftl1 mRNA levels were similar in WT and *Irp2*^{-/-} islets, but, unexpectedly, Fth1 mRNA levels decreased, which may..."

4. Figure 5a is reported to show a decrease in Tfr1 in cells treated with shIrp2 but that is not apparent in the second lane of the western blot. Could the results be quantitated, or is there a better blot?

Response: A new western blot (Fig. 5a) is included that better demonstrates reduced Tfr1 levels in shIrp2 cells.

Nancy Andrews

Reviewer #2 (Remarks to the Author):

The present study investigated the role of *Irp2* in the regulation of insulin biosynthesis and glucose metabolism. *Irp2*-knockout mice exhibited glucose intolerance due to an impairment in insulin secretion from pancreatic beta-cells. The decrease of insulin secretion was resulted from an increase in the proinsulin content and the decrease in the total insulin content in the beta-cells. Mechanistically, *Irp2*-deficiency suppressed the mRNA and protein levels of transferrin receptor (TfR1) and upregulated the protein level of ferritin, leading to the decrease in the total iron content in the pancreatic beta-cells. The decrease of intracellular iron subsequently impaired the activity of Cdkal1, an iron-sulfur cluster-dependent enzyme required for the methylthio-modification of tRNA^{Lys}(UUU). Dysregulation of Cdkal1 then impaired proinsulin translation at Lys codon. Importantly, supplement of iron in *Irp2*-deficient cells enhanced Cdkal1 activity, which improved the proinsulin translation fidelity and increased insulin contents. These results demonstrate that *Irp2* is essential for the iron homeostasis and controls translational fidelity of proinsulin via tRNA modification enzyme Cdkal1. Overall, these findings are interesting and novel.

Major concerns:

1. The 5-month old *Irp2*-knockout mice showed glucose intolerance after intraperitoneal (i.p.) injection of glucose (Figure 1a), but the plasma insulin levels of knockout mice did not differ from wild-type mice at the basal level and after the i.p. injection of glucose (Figure 2a). The plasma insulin data was not consistent with the hyperglycemic clamp data (Figure 2d), which showed that insulin secretion was markedly decreased in *Irp2*-knockout mice. Authors should explain this discrepancy.

Response: We repeated plasma insulin measurements in 7-month old WT and *Irp2*^{-/-} mice after an imp. injection of glucose that showed blunted insulin secretion in *Irp2*^{-/-} mice (Fig. 2a). In this experiments, plasma insulin levels were measured by a Mouse Insulin ELISA. The discrepancy in the original data is likely due to the use of an RIA KIT to measure plasma insulin, which we no longer use.

2. The finding that *Irp2*-deficiency resulted in the accumulation of proinsulin is particularly important in this study. Authors should examine the plasma proinsulin level or plasma proinsulin/insulin ratio in *Irp2*-deficient mice with imp. administration of glucose. Authors also need to examine the proinsulin secretion in isolated islets after glucose stimulation.

Response: The plasma proinsulin/insulin ratio was quantified in 7.5-month old WT and *Irp2*^{-/-} mice after an i.p. glucose injection using a proinsulin ELISA (Fig. 3d) and showed increased P/I ratio in *Irp2*^{-/-} mice vs WT. We also measured proinsulin secretion in WT and *Irp2*^{-/-} islets from 7.5 months old mice that shows increased proinsulin secretion in *Irp2*^{-/-} islets under basal glucose and after stimulation with high glucose (Fig. 3l and n).

Minor concerns:

1. Figure 3e and 3f. Authors showed that the average islet area and the beta-cell mass are comparable between *Irp2*-knockout mice and wild-type mice. HE staining of pancreas and islets need to be presented to support these data.

Response: H&E staining of representative pancreas sections from 7.5-month WT and *Irp2*^{-/-} mice is shown in Supplemental Fig. 3a-b. We also included representative insulin-stained pancreatic sections isolated from 7.5-month WT and *Irp2*^{-/-} mice that were used to measure islet area and mass (Fig. 3d-e).

2. Figure 3. There are two “b” panels. The upper right panel should be panel “c”.

Response: Corrected

3. Figure 4d. Authors claimed that TfR1 staining was reduced in *Irp2*-knockout beta-cells (page 7, line 164). This is not convincing because the quality of the TfR1 staining is very poor. A majority of beta-cells was not stained by the antibody even in the wild-type islets.

Response: We repeated immunostaining of our paraffin-embedded pancreatic sections using two other commercial TfR1 antibodies, but the images were not any better than the one in our manuscript. In addition to the antibodies that are not ideal, TfR1 is expressed at much lower levels than insulin, which may affect the image quality. Our graphic designer tried to obtain a better image, which is in the manuscript.

4. Figure 5a. Authors claimed that TfR1 was modestly reduced in *Irp2*-knockdown INS cells (page 8, line 175). This claim is not correct because the intensity of TfR1 band in *Irp2* knockdown cells was comparable to the TfR1 band in control cells (lane 1 versus lane 2). Please do not overstate the result, or author should provide the quantitative data.

Response: A new western blot (Fig. 5a) is included that better demonstrates reduced TfR1 levels in sh*Irp2* cells.

5. Figure 5b. Please provide the method for this experiment.

Response: A sentence to explain this method is included in the legend of Fig. 5b. “Whole cell lysates from EV and sh*Irp2* cells were incubated with a ³²P-labeled ferritin IRE RNA probe followed by the resolution of the *Irp1*- and *Irp2*-IRE complexes by non-denaturing polyacrylamide gels”. A more complete description of the method is included in Materials and Methods (RNA-electrophoretic mobility shift assays (RNA-EMSA)).

6. Figure 5c. In addition to Calcein-AM, authors need to use ICP-MS to measure the iron content in *Irp2*-knockdown INS cells (figure 5) and *Irp*-knockout INS cells (figure 6).

Response: As suggested, ICP-MS to measure total iron content in EV and sh*Irp2* cells (Fig. 5b) and in sg*Irp2.1* and sg*Irp2.2* INS-1 cells (Fig. 6b) that shows reduced total iron content in *Irp2*-deficient cells compared to control cells. The calcein-AM experiment was performed because at that time we did not have access to ICP-MS. The calcein-AM experiment was deleted.

7. Page 7, line 145: It is not appropriate to state that there is no general defect in ER protein maturation just because Glut2 showed no change between *Irp2*-knockout islets and wild-type islets. Please rephrase.

Response: Rephrased: “Insulin, glucagon and glucose transporter Glut2 immunostaining displayed normal subcellular localization, showing islet morphology is normal in *Irp2*^{-/-} islets (Supplementary Fig. 4a-b)”

8. Page 7, line 159: “... *Fth1* levels decreased in *Irp2*^{-/-} islets...”. Should this be “... *Fth1* levels increased in *Irp2*^{-/-} islets...”?

Response: Yes, and corrected. “*Fth1* was similarly expressed in WT and *Irp2*^{-/-} islets, while *Fth1* was not detected in WT islets, but was notably increased in *Irp2*^{-/-} islets (Fig. 4a).”

9. Page 7, line 162: “... *Fth1* mRNA levels increased...”. Should this be “... *Fth1* mRNA levels decreased...”?

Response: Yes, and corrected. “*Fth1* mRNA levels were similar in WT and *Irp2*^{-/-} islets, but, unexpectedly, *Fth1* mRNA levels decreased, which may be a transcriptional...”

10. Page 8, line 189: “...from reduced insulin content...”. Should this be “...from increased insulin content...”?

Response: We reworded this sentence: “When the amount of secreted insulin was normalized to insulin content, sh*Irp2* cells secreted slightly less insulin compared to EV cells, suggesting that secretion may be somewhat impaired (Fig. 5h).”

11. Page 13, line 362. “min 120” should be “120 min”.

Response: Corrected

12. The units for glucose and insulin are not consistent. For example, insulin is shown as ng/ml in figure 2b, but is shown as pmol/l in figure 2d. Please be consistent.

Response: Corrected - Insulin units changed to ng/ml in Fig. 2d.

13. Please provide sequencing data to show how *Irp2* gene was edited in the two lines of *Irp2*-knockout cells

Response: CRISPR-edited *Irp2* cells lines were generated using two distinct sgRNAs sgIrp2.1 (against exon 3) and sgIrp2.2 (against exon5). Crispr-*Irp2* KO cell pools were selected using puromycin and assayed for loss of *Irp2* by western blotting. Because our two *Irp2* Crispr-*Irp2* cell lines represent mixed pools, they were not sequenced.

Reviewer #3 Specific Comments:

The main point of this paper is that beta cell deficiency of labile iron perturbs the fidelity of proinsulin translation leading to diabetes. Specifically, in this manuscript, the authors utilize mice with whole body knockout of the RNA-binding protein *Irp2*, and show that these mice develop glucose intolerance with an increase in intracellular proinsulin and a decrease in intracellular insulin. They indicate that *Irp2* loss of function leads to increased sequestered iron but decreased labile iron that they link with destabilization of the Fe-S enzyme *Cdkal1* (itself a T2D susceptibility gene), thus resulting in improper translation of lysine codons in proinsulin that could account for impaired proinsulin processing and possible beta cell ER stress. Using *Irp2* deficient INS1 cells as a model, some of these adverse effects are improved by preincubation for 18 hours with ferric ammonium citrate (FAC) to increase labile iron. The connection of *Irp2* levels to regulation of *Cdkal1* to ultimately drive regulation of proinsulin processing through methylthiolation and proper lysine codon translation is novel, and the story is well written. However, a major concern is insufficient validation and the lack of more physiological evidence to support the mechanism that they are hypothesizing. At this point, additional studies are necessary to support the conclusions and enhance the significance of the manuscript.

Specific Comments:

1. The authors show that 18 hours of FAC supplementation rescues insulin-related phenotypes in *Irp2*-deficient beta cells. To demonstrate physiological significance, the authors should try to determine if iron supplementation (even a relatively brief course to prevent toxicity) can restore insulin production or secretion in vivo.

Response: We agree and carried out an in vivo supplementation experiment using WT and *Irp2*^{-/-} mice. Mice were intraperitoneally-injected with iron dextran for 5 consecutive days, after which mice were sacrificed and pancreata harvested for insulin and proinsulin determination by ELISA, and iron content by ICP-MS. The data show that iron supplementation partially normalizes iron content and fully restores proinsulin iron content in *Irp2*^{-/-} mice. Iron overload in mice was verified by ICP-MS analysis of iron content in liver and pancreas. These experiments are described in Fig. 5j-m.

2. Independent of #1, the authors should determine if FAC is capable to restore insulin processing and *Cdkal1* levels in isolated *Irp2* null islets instead of relying solely on results from celllines.

Response: Our plan was to carry out a FAC experiment on isolated islets. Unfortunately, our supply of *Irp2*^{-/-} mice was limited and mice were used for proinsulin secretion experiments suggested by Reviewer 2. We did not obtain enough islets from a mouse to perform proinsulin and insulin secretion experiments plus and minus iron. In lieu of using isolated islets, we performed the in vivo experiment

suggested in Comment 1.

3. Immunofluorescence microscopy with co-stained organelle markers, and transmission electron microscopy of the islets of *Irp2* null pancreata are needed to understand where in beta cells the increased proinsulin may be accumulating.

Response: Several studies have reported perinuclear staining-Golgi like staining for proinsulin (Zhu et al. PNAS 2002 and Haataja, L, et al. PNAS, 2013). Under conditions where proinsulin accumulates due to misfolding and/or miscleavage, proinsulin can be found in the ER (Haataja, et al. JBC 288:1896, 2013) and in large aggregates in the cytoplasm (Back et al, Cell Met, 2009; Zhu et al., PNAS 2002; Wei et al., JCI, 2011). We assessed proinsulin localization staining in WT and *Irp2*^{-/-} islets, but we did not observe significant differences in between WT and *Irp2*^{-/-} except for increased PI staining in *Irp2*^{-/-} islets (attached figure), and therefore, we did not include these studies in the manuscript.

4. Many of the mechanistic connections drawn between *Irp2* and impaired insulin processing may be linked by effects on *Cdkal1*, but some attempts should be made to exclude other possibilities.
 - a. Fe-S clusters regulate key mitochondrial proteins and proinsulin intracellular transport and processing is an energy intensive process. Low Complex I activity in sh*Irp2* INS-1 cells could lead to the generation of ROS, which could then lead to untoward downstream effects. Thus, pharmacologic approaches to enhance mitochondrial fuel utilization (such as mitochondrial targeted substrates including methylpyruvate or amino acids leucine/glutamine/alanine) and/or reduce ROS (antioxidants) independent of using an iron supplement are needed to determine if it is truly *Cdkal1* deficiency that impairs insulin processing in *Irp2* null islets or simply bioenergetic failure.

Response: We also thought that reduced Complex 1 activity in *Irp2*-deficient cells would lead to increased ROS production. We found that ROS were not significantly altered in sh*Irp2* INS-1 vs control cells; however, a slight increase in ROS levels was observed in sh*Irp2* INS-1 cells vs EV cells after a short exposure to H₂O₂ (see figure below), suggesting that *Irp2*-deficient cells are sensitive to oxidative stress. We also did not find altered expression of anti-oxidant enzymes in *Irp2*-deficient cells.

[redacted]

Similar to β -cell specific *Cdkal1* KO mice (Wei, et al. JCI, 2011), we found reduced ATP generation after glucose stimulation and reduced first phase-insulin secretion. How *Cdkal1* loss affects mitochondrial function is not yet fully understood. Wei et al. suggested that *Cdkal1* does not directly affect mitochondrial function, but might regulate the translation of proteins involved in mitochondrial function. While we cannot completely exclude a mitochondrial bioenergetic failure contributing to aberrant PI processing in *Irp2*-deficient cells, our data showing reduced *Cdkal1* protein and activity (measured by western blots and ms²t⁶A levels), reduced lysine incorporation into proinsulin and restoration of *Cdkal1* function by iron strongly suggests that *Cdkal1* deficiency in *Irp2*-deficient cells is responsible for impaired proinsulin processing. We are interested in the mitochondrial phenotype in *Irp2*-deficient cells and are working on this project, but feel that these experiments are beyond the scope of this manuscript.

- b. Can overexpression of Cdkal1 more efficiently rescue proinsulin processing deficits in Irp2 null cells plus and minus FAC?

Response: We considered Cdkal1 overexpression, but given reduced Fe-S cluster biosynthesis and iron deficiency in Irp2-deficient cells, it is likely that ectopically expressed Cdkal1 would not be active. We also considered overexpression plus iron, but were concerned that increased Cdkal1 function may have untoward effects on cells (e.g. methylthiolation of other substrates) that might confound our studies. We found that iron supplementation alone in Irp2-deficient cells was sufficient to normalize Cdkal1 levels and activity, and lysine incorporation into PI.

3. As best I can tell, the Irp2 null mice have been around for 15 years, but implicating deficiency of Irp2 directly in human disease has not really happened yet. As far as I know, Irp2 is not even on the long list of diabetes GWAS candidates. With this in mind, it would be best if the authors would attempt to demonstrate a proinsulin processing defect in primary human islets with deficiency of labile iron, perhaps by utilizing iron chelation on the isolated islets.

Response: Recently, two patients have been identified with mutations in *IREB2*. Both patients exhibit early onset and progressive neurological disease and microcytic anemia (Cooper et al. BRAIN, 2019 and Costain, et al. BRAIN 2019). Glucose studies were not reported for these patients, and unfortunately, one patient died and the other patient is no longer participating in research studies. Humans with *IREB2* mutations are rare, but if new patients are identified, it would be of interest to determine if glucose metabolism is altered in these patients. What our data suggests is a link between iron deficiency, which is common iron disorder, and insulin processing that may be relevant to humans.

We agree that studying proinsulin processing in context of iron deficiency in primary human islets is good idea. For these studies we need to use several batches of human islets, and will take us several months to carry out these experiments (e.g. knockdown of Irp2, iron chelation and iron supplementation studies), and will not be feasible to include these studies in this manuscript.

4. The activation of eIF2 α may reflect ER stress; additionally iron status might independently affect eIF2 α phosphorylation – these points are poorly developed. At minimum, the authors should re- check phospho-eIF2 α levels in IRP2 deficient cells treated \pm FAC, as well as a similar experiment after Cdkal1 overexpression.

Response: Iron status has been shown to increase ER stress. For example, iron chelation in insulinoma cells increased the levels of phospho-eIF2 α (Jung, I. et al. Mol Cell Endo, 2015) and reduced glucose-stimulated insulin secretion, which is consistent with our studies. Because of reasons discussed above regarding Cdkal1 overexpression, these experiments were not performed.

Minor:

5. The Figure 6G legend was omitted.

Response: Corrected. “Western blot analysis of eIF2 α -P, eIF2 α -total and Grp78/BiP levels in control, sglrp2.1 and sglrp2.2 cells under basal glucose (5 mM) and after stimulation with 15 mM glucose”.

6. The FAC abbreviation on line 473 is misplaced in the sentence.

Response: Corrected

Reviewers' comments:

Reviewer #1 (Remarks to the Author):

My points have been addressed.

Reviewer #2 (Remarks to the Author):

Authors have fully addressed my concerns. I have no further comments.

Reviewer #3 (Remarks to the Author):

The authors did a nice job of in vivo Fe supplementation in response to the previous query. On the other hand, they did not do the experiment of adding FAC to isolated islets, which would test the beta cell more directly; and they did not attempt any experiment with human islets.

I do not wish to make new suggestions, as this is the second round of review.

I do recommend that the authors consider that the section in the Results on ER stress follow the section on cdkal1 deficiency, and that both the Results and Discussion point out that the 2011 JCI paper by Wei, cited by the authors, has already reported that cdkal1 deficiency results in induction of ER stress.

Response to reviewers' comments:

We appreciate the thorough critique of our manuscript that we found insightful and constructive. Based on the reviewers' comments and suggestions, we have revised the manuscript. New experiments and responses to reviewers' comments are described below.

Editors' Comment: In particular, after considering Referee 3's advice and discussion with the editorial team we think that the referee's request for additional experiments to confirm the effects of iron directly on mouse islets should be addressed.

Reviewer 1:

My points have been addressed.

Reviewer 2:

Authors have fully addressed my concerns. I have no further comments.

Reviewer 3:

1. The authors did a nice job of in vivo Fe supplementation in response to the previous query. On the other hand, they did not do the experiment of adding FAC to isolated islets, which would test the beta cell more directly; and they did not attempt any experiment with human islets.

Response: We performed iron and iron chelator experiments in isolated islets from WT and *Irp2*^{-/-} mice. Islets were isolated from 7.5-month old mice (n=7-8 mice per genotype) and cultured overnight in medium supplemented with ferric ammonium citrate (iron, FAC) or the iron chelator desferrioxamine (DFO). Insulin and proinsulin secretion and content were then assayed under conditions of high glucose. Our data show that iron normalizes proinsulin and insulin secretion, and content in *Irp2*^{-/-} islets, while DFO reduces insulin secretion and content, and increases proinsulin secretion and content in WT islets. These data are consistent with iron and DFO studies carried out in INS-1 832/13 cells (now Fig. 6) and in *Irp2*^{-/-} mice injected with iron dextran (Fig. 5f-i). Please note that these studies are now presented in new **Fig. 5a-e** (highlighted) along with the *Irp2*^{-/-} in vivo iron study (**Fig. 5f-i**). Also, the number of figures increased from 7 to 8 to accommodate new data.

2. I do not wish to make new suggestions, as this is the second round of review.

3. I do recommend that the authors consider that the section in the Results on ER stress follow the section on cdkal1 deficiency, and that both the Results and Discussion point out that the 2011 JCI paper by Wei, cited by the authors, has already reported that cdkal1 deficiency results in induction of ER stress.

Response: Wei, JCI 2011 was cited in the Discussion with reference to ER stress in β -cell specific Cdkal1KO mice.